# Angiogenesis is uncoupled from osteogenesis during calvarial bone regeneration

M. Gabriele Bixel [1] ✉, Kishor K. Sivaraj [1], Melanie Timmen [2], Vishal Mohanakrishnan [1], Anusha Aravamudhan [1], Susanne Adams[1], Bong-Ihn Koh [1], Hyun-Woo Jeong [1,3], Kai Kruse [1,4], Richard Stange [2] & Ralf H. Adams [1] ✉

Bone regeneration requires a well-orchestrated cellular and molecular response including robust vascularization and recruitment of mesenchymal and osteogenic cells. In femoral fractures, angiogenesis and osteogenesis are closely coupled during the complex healing process. Here, we show with advanced longitudinal intravital multiphoton microscopy that early vascular sprouting is not directly coupled to osteoprogenitor invasion during calvarial bone regeneration. Early osteoprogenitors emerging from the periosteum give rise to bone-forming osteoblasts at the injured calvarial bone edge. Microvessels growing inside the lesions are not associated with osteoprogenitors. Subsequently, osteogenic cells collectively invade the vascularized and perfused lesion as a multicellular layer, thereby advancing regenerative ossification. Vascular sprouting and remodeling result in dynamic blood flow alterations to accommodate the growing bone. Single cell profiling of injured calvarial bones demonstrates mesenchymal stromal cell heterogeneity comparable to femoral fractures with increase in cell types promoting bone regeneration. Expression of angiogenesis and hypoxia-related genes are slightly elevated reflecting ossification of a vascularized lesion site. Endothelial Notch and VEGF signaling alter vascular growth in calvarial bone repair without affecting the ossification progress. Our findings may have clinical implications for bone regeneration and bioengineering approaches.

Despite substantial advancements in our understanding of the mechanisms underlying bone regeneration, major challenges in orthopedic surgery remain due to failing or delayed fracture healing and complications during bone repair[1,2]. Segmental bone defects caused by trauma, infections and tumors associated with insufficient osteogenesis often result in significant disabilities in patients[3–5]. The reasons for failed bone regeneration and non-union fractures often remain unclear[6,7]. Bone repair is achieved through a highly complex and interconnected series of cellular and molecular events orchestrated by various mediators and signaling factors. In long bone, fracture healing involves several distinct stages starting with early hematoma formation, followed by a reparative and remodeling

[1]Department of Tissue Morphogenesis, Max Planck Institute for Molecular Biomedicine and University of Münster, Faculty of Medicine, D-48149 Münster, Germany. [2]Department of Regenerative Musculoskeletal Medicine, Institute of Musculoskeletal Medicine, University Hospital Münster, D-48149 Münster, Germany. [3]Max Planck Institute for Molecular Biomedicine, Sequencing Core Facility, D-48149 Münster, Germany. [4]Max Planck Institute for Molecular Biomedicine, Bioinformatics Service Unit, D-48149 Münster, Germany. ✉e-mail: mgbixel@mpi-muenster.mpg.de; ralf.adams@mpi-muenster.mpg.de

response[8–10]. Critical steps in the progression of bone repair are robust vascularization and recruitment of mesenchymal and osteogenic cells to the site of injury[8,11,12]. Vascularization and osteogenesis are closely coupled during this process through specialized blood vessels that provide paracrine signals to coordinate osteoprogenitor migration and differentiation in bone development and regeneration[13,14]. The signaling interactions that regulate co-invasion of osteoprogenitors with blood vessels during fracture repair are still poorly understood[15]. Candidate mediators involved in osteo-angiogenic coupling are components of the vascular endothelial growth factor (VEGF) and hypoxia inducible factor (HIF) pathways, Notch and, more recently, platelet derived growth factor receptor beta (PDGFRβ) signaling[13,16,17]. PDGFRβ has been suggested to be a critical functional driver for mesenchymal skeletal stem cell activation, migration and angiotropism during bone repair[18]. In the initial phase, the hematoma creates a special milieu of signaling factors, including chemotactic factors and cytokines, to attract inflammatory cells[10,19]. While macrophages clear the necrotic tissue and fibrin degradation products[10,20], inflammatory cells release chemotactic and growth factors to promote the recruitment of various cell types including vascular endothelial cells, mesenchymal stromal cells (MSCs) and fibroblasts. At the fracture site, MSCs proliferate and differentiate in close proximity to blood vessels into osteoprogenitors and osteoblasts to promote bone regeneration[19,21,22]. Long bone fractures typically involve interfragmentary strain that induces chondroid soft callus formation consisting of strain-absorbing extracellular matrix[23]. Hypertrophic chondrocytes secrete cytokines and growth factors to attract vascular endothelial cells and osteoprogenitors. Sprouting capillaries originate from the periosteal and intramedullary vasculature[24,25] and infiltrate the avascular callus in close association with osteoprogenitors to promote callus remodeling and endochondral ossification[10,23,26]. HIF signaling, strain and vascularization are absolutely essential for bone repair[10,12] and dictate the type of ossification that occurs, either by direct intramembranous ossification and/or by endochondral ossification involving a cartilage intermediate[10,27]. Small, stabilized fractures, i.e. drill hole lesion injuries in femur[28,29] or calvarium[30] heal in absence of interfragmentary strain by intramembranous ossification. Calvarial lesion injuries have been used previously to study vascularization and ossification[31,32], the oxygen microenvironment[31] and bioactive tissue transplants[33,34] in bone regeneration. However, the highly dynamic and complex cellular interactions including cell proliferation, migration and differentiation and their timely coordination and regulation during calvarial bone healing still remain poorly understood.

In recent decades, major developments in imaging technologies, in particular confocal and multiphoton microscopy as well as micro-computed tomography, have significantly improved our understanding of the microarchitecture and specialized functions of various tissues, including the bone and bone marrow (BM) environment[14,35,36]. In mice, intravital multiphoton imaging of the calvarium and long bones have been used to visualize trafficking and dynamic behavior of cells in the BM[37–40], blood flow dynamics in BM microvessels[41], vascular plasticity[42] and bone regeneration[31,43].

The study of regenerative processes in the bone using intravital multiphoton microscopy is challenging, since bone is an optically dense and highly scattering tissue. Here, we present a greatly improved methodology of longitudinal intravital imaging, which allows to visualize repeatedly in a single living animal the highly complex process of calvarial bone healing after lesion injury at a multiscale level until lesion closure using high-resolution multiphoton microscopy. This advanced imaging approach enables us to visualize and track how i) regenerating blood vessels vascularize the calvarial bone lesion and establish a mature and perfused vascular network, ii) vascular sprouts and early osteoprogenitors migrate into the bone lesion in a spatiotemporal fashion, iii) bone-forming osteoblasts initiate and progress the formation of new bone until

lesion closure, and iv) activation and inhibition of endothelial Notch, and enhanced VEGF-A signaling affect blood vessel sprouting and remodeling without impacting ossification of the healing bone. Furthermore, we compare calvarial lesions with femoral fractures with regard to vascularization and association with osteoblastic cells and identify striking differences in osteo-angiogenic coupling during bone repair. Single-cell RNA sequencing (scRNA-seq) results provide insight into bone stromal cell heterogeneity and expression of genes related to angiogenesis and oxygen sensing during bone regeneration. The full scRNA-seq data is accessible through an online cell viewer platform allowing easy and rapid interrogation of results.

## Results
### Vascular regeneration and calvarial bone healing after lesion injury

To gain insight into the dynamic vascular changes after calvarial bone injury, we generated drill hole lesions in the parietal skull bone of *Flk1-GFP*⁺ reporter mice. The labeling of ECs allows the monitoring of vascular growth and remodeling processes in the lesion using longitudinal intravital multiphoton microscopy (Fig. 1a–c). An optical cranial window was mounted on the injured bone and imaged in intervals of typically three days over a period of up to six weeks. After an initial avascular phase, first *Flk1-GFP*⁺ endothelial sprouts emerge from the underlying vascular network around post-lesion day (PLD) 6 and show a characteristic tip cell morphology with filopodial extensions (Fig. 1c). Whole mount analysis of the calvarium at PLD3 and 6 reveals that early *Flk1-GFP*⁺ sprouts originate from the meningeal vascular network (Supplementary Fig. 1a–e). Second-harmonic generation (SHG) imaging allows simultaneous visualization of *Flk1-GFP*⁺ vessels and calcified collagen network in the bone tissue. Bone lesions appear as circular dark areas, surrounded by the adjacent SHG⁺ calvarial bone (Fig. 1c). In the following days, profound and robust sprouting angiogenesis results in vascularization of 34 ± 3% of the defect area by PLD9 (Fig. 1c, e). Then, outward growing vascular sprouts emerge from the early vascular network and connect with outer periosteal vessels around PLD9-12 (Fig. 1c, Supplementary Fig. 1e, f). At the same time, *Flk1-GFP*⁺ microvessels closely align with the surface of the adjacent injured calvarial bone (Supplementary Fig. 1f). These bone-aligned microvessels become part of a dense vascular network that covers the complete lesion area. Repetitive imaging of selected areas allows tracking of the dynamic vessel sprouting and remodeling processes in the lesion and reconstruction of the resulting vascular changes (Supplementary Fig. 2a). Endothelial sprouting generates new connections between existing microvessels and thereby establishes new vascular branch points. In the same period, nearby EC tubes are subjected to pruning, leading to the loss of vascular connections (Supplementary Fig. 2b). Robust and highly dynamic vascular sprouting is followed by a phase of extensive vascular remodeling during which the newly formed vasculature is embedded in the healing bone (Fig. 1d). This process continues in the following weeks and significantly reduces vascular density to reach a vascular area of 20 ± 1% in the lesion around 3-weeks post-lesion (Fig. 1e, f).

Deposition of new SHG⁺ bone matrix is first observed around PLD9 at the injured edge of the calvarial bone (Supplementary Fig. 2c). The ossification process advances with new bone steadily growing inwards from the adjacent ossified areas, resulting in lesion closure over a period of 4-5 weeks (Fig. 1d, e). The velocity of local bone formation varies considerably in the range of 249 ± 154 μm²/h, however, averaged across the whole lesion, growth is steady during the observation period (Fig. 1e, Supplementary Fig. 2c). Collagen fibers of the new bone matrix assemble a dense and interconnected three-dimensional network with an average fiber diameter of 2.2 ± 0.2 μm at the growing front (Fig. 1d, Supplementary Fig. 2d). *Flk1-GFP*⁺ microvessels are located in close proximity to these collagen fibers and are frequently surrounded by growing bone, which leads to the

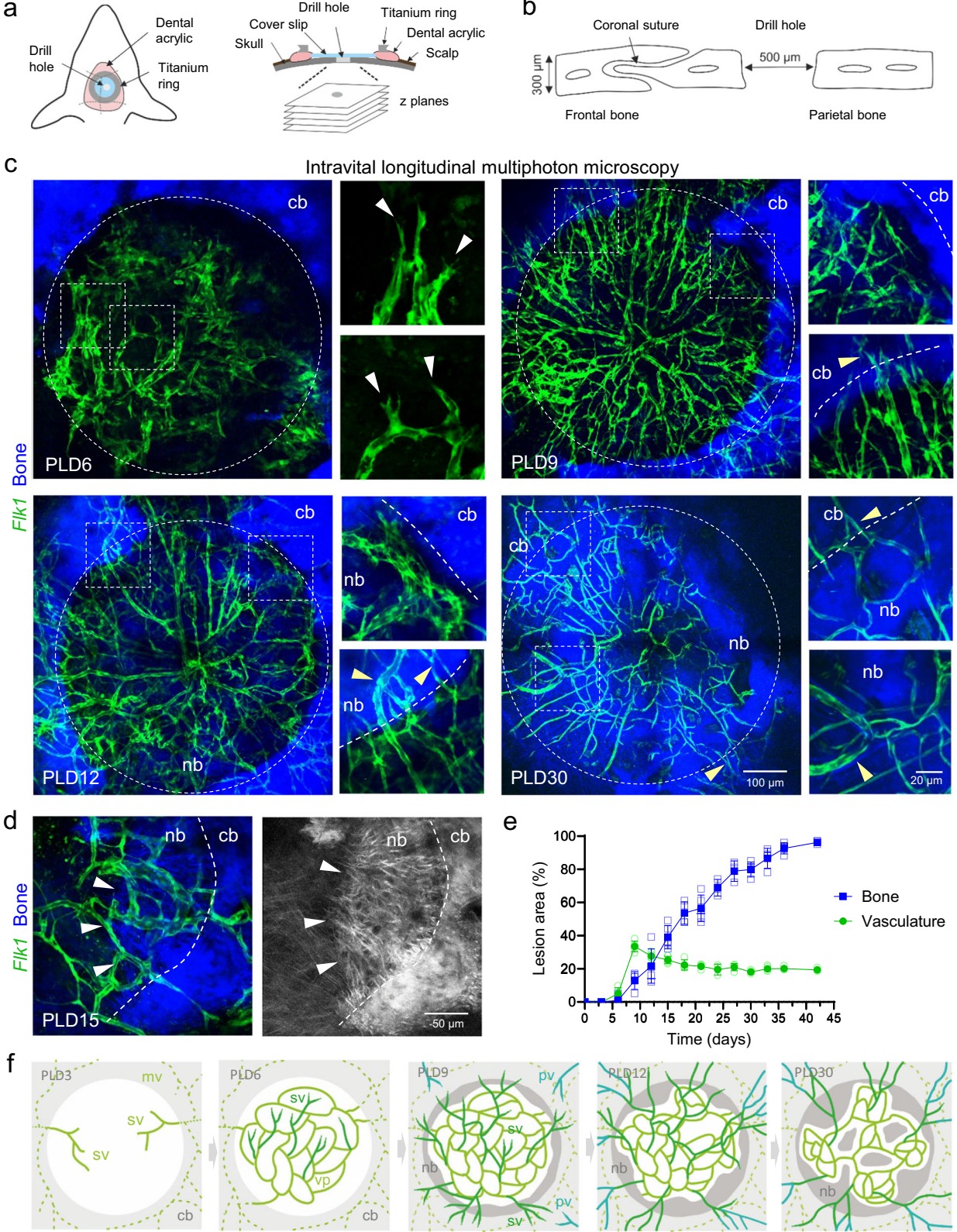

formation of early bone marrow-like cavities (Fig. 1d, f). The SHG⁺ fiber network in the uncalcified lesion is clearly distinct from the growing bone. These collagen fibers are thinner and show a parallel alignment (Supplementary Fig. 2c, d). *Flk1-GFP⁺* vascular sprouts are typically in close proximity or attached to SHG⁺ fibers, indicating that endothelial cells might extend along these collagen fiber scaffold, as reported previously[44] (Supplementary Fig. 2d). Taken together, these results provide a spatiotemporal framework for vascular growth during bone generation in the calvarium.

### Vascular connectivity and blood flow in regenerating bone

Intravital multiphoton imaging shows that the regenerating bone in the lesion, with its growing and enclosed vasculature, is closely aligned with the adjacent calvarial bone of the former wound edge (Fig. 2a,

**Fig. 1 | Longitudinal intravital multiphoton imaging of calvarial bone healing after lesion injury. a** Schematic of a chronic cranial window for intravital imaging of calvarial bone repair. A drill hole lesion is inserted in the parietal calvarial bone. Coverslip and titanium fixation ring are mounted over the lesion. The lesion area of the is visualized using multiphoton laser microscopy. Modified from Stewen J, Bixel MG. Intravital Imaging of Blood Flow and HSPC Homing in Bone Marrow Microvessels. Methods Mol Biol. 2019;2017:109-121[84] © 2019 Springer Science+Business Media, LLC, part of Springer Nature. **b** Schematic showing a drill hole lesion the parietal calvarial bone. **c** Longitudinal intravital multiphoton microscopy showing dynamic sprouting and remodeling of GFP+ (green) microvasculature and regenerating SHG+ (blue) bone matrix in *Flk1-GFP* reporter mice after calvarial bone injury. Overview images show maximum intensity projections of *Flk1-GFP+* microvasculature, injured SHG+ calvarial bone (cb) and newly formed SHG+ bone matrix (nb). Zoom-in views show early *Flk1-GFP+* sprouts (white arrow heads) at PLD6, *Flk1-GFP+* microvessels aligning the injured calvarial bone (top) and early sprouts (yellow

arrow head, bottom) reaching over the outer bone edge at PLD9, remodeled *Flk1-GFP+* microvessels connecting to periosteal vessels (yellow arrow heads) at PLD12, and matured *Flk1-GFP+* microvessels enclosed by new SHG+ bone matrix at PLD30. **d** Intravital multiphoton microscopy showing SHG+ fibers of new bone (nb) originating from the injured calvarial bone (cb) in close association with *Flk1-GFP+* blood vessels at PLD15. Arrow heads point to the front of the growing bone. **e** Histogram showing new microvasculature and new bone in the lesion (in % of total lesion area). Data are presented as mean values ± SD with $n = 5$ (for PLD0-PLD21), $n = 4$ (for PLD24-33) and $n = 3$ (for PLD36-42) biologically independent animals. Source data are provided as a Source Data file. **f** Schematic showing various stages of calvarial bone repair. New vascular sprouts (sv) originate from the meningeal vasculature (mv) and form an early vascular plexus (vp) that connects with the periosteal vasculature (pv). Formation of new bone (nb) initiates at the injured calvarial bone surface (cb) and progressively grows into the vascularized lesion gradually remodeling or enclosing blood vessels.

Supplementary Movie 1). However, analysis of sagittal sections shows that the vasculature in the lesion and vessels in the adjacent uninjured bone and marrow do not have direct vascular connections, possibly due to a physical barrier created by the separating bone (Fig. 2b, c). Instead, there are many vascular connections between the lesion and the outer periosteal and inner meningeal vasculature (Fig. 2e, see below, Supplementary Fig. 1c-f). Unexpectedly, the newly formed bone attaches and connects only irregularly to the adjacent injured calvarial bone, often leaving a visible gap for several weeks (Fig. 2d). Bundles of collagen fibers cut by the drilling process often remain blunt-ended (Fig. 2b, d). While collagen fibers are preferentially located in densely packed bundles parallel to the surface of the uninjured calvarial bone (Fig. 2b, c), fibers in the newly formed bone are substantially less organized and arranged in an irregular crisscross pattern (Fig. 2b, Supplementary Fig. 2d, Supplementary Movie 2).

To understand how blood flow is established in the expanding vascular network during calvarial bone healing, we performed direct in vivo measurements of red blood cell (RBC) movement using real-time movies and centerline scans to determine RBC velocities at the level of individual vessel segments[41]. Intravenous injection of fluorescently labeled dextran visualizes lumen-containing vascular sprouts at PLD7 that emerge from the early vascular plexus and extend outwards towards the periosteal vasculature (Fig. 3a). Filopodia decorate the tip of these vascular sprouts, while trailing stalk cells enclose an early vascular lumen connecting to previously established parts of the vessel network (Fig. 3a). Growing sprouts are often filled with stacks of RCBs trapped in blind-ending luminal spaces that are largely stationary and move only passively due to blood circulation in neighboring vessel segments, as visualized by real-time imaging (Supplementary Movie 3, 4). Repetitive centerline scans along individual vessel segments allowed the assembly of a flow map showing the range of RBC velocities as well as perfused and non-perfused luminal spaces (Fig. 3b). In the early expanding vascular network at PLD7, RBC flow velocities are highly variable among individual vessels independent of their diameter, which is likely to reflect dynamic changes in vascular connectivity within the remodeling network (Fig. 3b, c).

In previous work, we had characterized hemodynamic parameters and vessel dimensions in different bone marrow microvessels[41]. In the current study, we measured blood flow velocities and vessel diameters at PLD9 and PLD21 and used the previously described flow parameters to categorize the expanding and remodeling vascular network as arterial vessels (>1.4 mm/sec), post-arterial capillaries (1.4-.0-8 mm/sec), intermediate vessels (0.8-0.5 mm/sec) and sinusoidal capillaries (< 0.5 mm/sec). At PLD9, the majority of vessels in the early vascular network shows a luminal diameter of 8.2 ± 2.2 µm irrespective of the measured heterogenous flow velocities (Fig. 3d). In the more mature vascular network at PLD21, RBC flow velocities decrease significantly with increasing luminal diameter and differentiation into early arterial

and post-arterial vessels of 10.8 ± 2.4 µm and 10.6 ± 2.5 µm, intermediate capillaries of 15.0 ± 5.9 µm and sinusoidal capillaries of 20.8 ± 8.7 µm (Fig. 3e). Microvessels in close proximity to growing bone are strategically located to allow gas exchange and transport nutrients and minerals to bone-forming osteoblasts lining the bone[13]. To characterize blood flow in these microvessels, we measured RBC velocities at PLD21 in selected vessels close to the edge of growing bone and at locations where vessels are partially enclosed by expanding bone (Fig. 3f, g). Surprisingly, RBC flow velocities are low (0.50 ± 0.16 mm/sec) in these microvessels with small luminal diameter (11.8 ± 16 µm).

This part of the work illustrates how growth and remodeling of the microvasculature lead to dynamic flow changes at the lesion site.

## Osteoprogenitors emerge from the periosteum to enter calvarial bone lesion

To gain insight into the early events after calvarial bone injury and the source of osteoprogenitors, identified by expressing the transcription factor Osterix (Osx, encoded by the gene *Sp7*), drill hole lesions in whole mount calvarial bones of *Flk1-GFP* and *Sp7-mCherry* double transgenic mice were analyzed at PLD4 and 6 (Fig. 4a). At this early stage, Osx+ osteoprogenitor cells with high mCherry expression are abundant in the periphery of the lesion area (Supplementary Fig. 3a–d). *Sp7-mCherry+* cells originate from the inner periosteal layer adjacent to the meningeal vasculature, which is consistent with previous reports[33] (Fig. 4a–d). *Sp7-mCherry+* cells increase in the periosteal layer and collectively migrate towards the lesion site (Fig. 4f, Supplementary Fig. 3f, g). The majority of periosteal *Sp7-mCherry+* cells infiltrate the lesion as a multicellular layer and align closely to the edge of the calvarial bone injury, a region that also contains microvessels emerging from the meningeal vasculature (Fig. 4a–f). A minor subset of *Sp7-mCherry+* cells with spindle-shape morphology appears to migrate as individual cells into the injury site, often in proximity to early microvessels (Fig. 4f, g). However, many of the microvessels in the lesion at a greater distance from the injured bone edge are not associated with *Sp7-mCherry+* osteoprogenitors (Fig. 4c–f, h).

Runx2 and Osx are transcription factors promoting different steps in osteogenesis[45–47]. While Runx2 is required at an early stage for the conversion of primitive mesenchymal cells into preosteoblasts, Osx drives the differentiation of these cells into mature osteoblasts[18,48]. Osx+ osteoprogenitors gradually differentiate into SHG+ bone-matrix secreting Osx+ osteoblasts (Fig. 4j). Immunostaining of *Flk1-GFP, Sp7-mCherry* calvarial bones shows that Runx2+ osteoprogenitors are abundant at PLD6 in the expanded periosteal layer adjacent to the lesion but also inside the injured area (Supplementary Fig. 3d, e). At this stage, early *Sp7-mCherry+* osteoblasts are confined to the periosteum and the surface of the wound edge (Supplementary Fig. 3d, e). While *Sp7-mCherry+* osteoblasts at the edge of the lesion are strongly

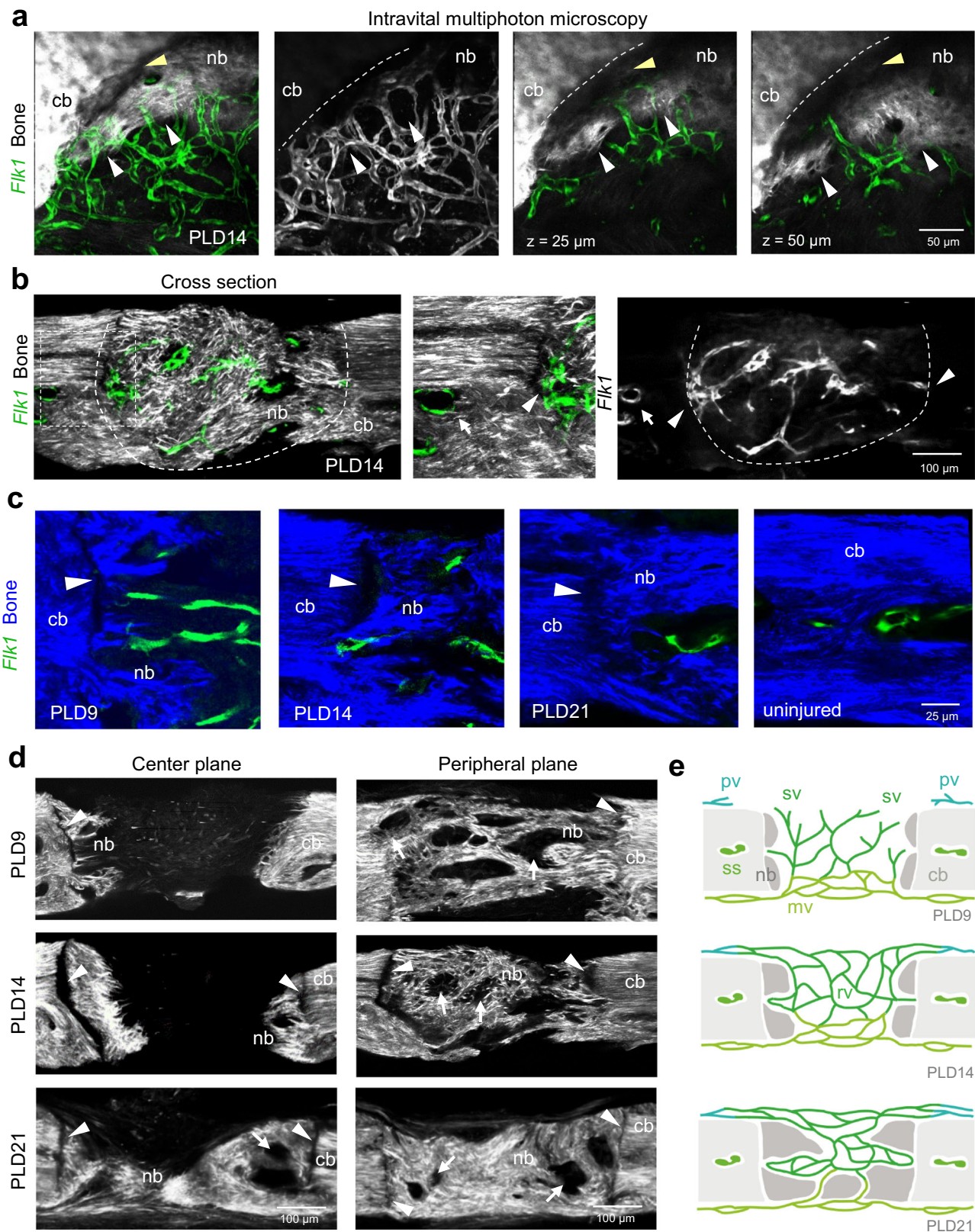

positive for *Sp7-mCherry* expression, Runx2⁺ cells with a spindle-shaped morphology and less bright *Sp7-mCherry* reporter signal are invading the lesion (Supplementary Fig. 3e, f). These cells are likely to represent early Osx⁺ Runx2⁺ osteoprogenitors and can be found in proximity of early *Flk1-GFP⁺* endothelial sprouts (Fig. 4h,

Supplementary Fig. 3e, f). By contrast, Osx⁺ Runx2⁺ double positive cells are absent in regions further away from the lesion area (Supplementary Fig. 3e). PDGFRβ signaling promotes a proliferative, immature and migratory status of skeletal stem and progenitor cells both in development but also fracture repair[18,49]. PDGFRβ immunostaining is

**Fig. 2 | Vascular and bone connectivity during calvarial bone healing.**
**a** Intravital multiphoton microscopy showing newly formed calvarial bone matrix and regenerating microvasculature at PLD14. Maximum intensity projections (left) of SHG+ (white) injured calvarial bone (cb), new SHG+ bone (nb) and *Flk1-GFP+* (green). White arrowheads indicate the growing bone edge. Separating gap (yellow arrowhead) between newly formed and pre-existing calvarial bone. Note that *Flk1-GFP+* microvessels are aligned with the surface of calvarial bone without connections to vessels inside the calcified tissue. Single planes on the right. Corresponding 3D representation is shown in Supplementary Movie 1. **b** Cross section showing *Flk1-GFP+* (green) microvasculature and SHG+ (white) calvarial bone at PLD14 (maximum intensity projection). Arrow points to sinusoidal capillaries in the adjacent BM cavity without vascular connections to the regenerating vasculature (arrowheads). **c** Cross sections showing *Flk1-GFP+* (green) microvasculature and SHG+ (blue) calvarial bone at PLD9, PLD14 and PLD21, and control without lesion. Arrowheads point to gaps separating injured calvarial bone (cb) and new bone (nb). **d** Overview cross sections showing SHG+ (white) calvarial bone with lesions at PLD9, PLD14 and PLD21. Lesions were cut centrally (left) and peripherally (right). Arrowheads point to gaps separating calvarial (cb) and new bone (nb). Arrows indicate new BM cavities. **e** Schematic cross sections at three stages of calvarial bone healing. Sprouting vessels (sv) derived from meningeal vessels (mv) form an early vascular plexus. Sprouts grow outward and connect to periosteal vessels (pv). The newly formed microvasculature remodels and adapts to the expanding new bone (nb). Progressive ossification results in lesion closure and formation of early BM cavities. cb: calvarial bone. Reproducibility was ensured by *n* = 3 or more biologically independent experiments.

highly abundant inside the lesion, suggesting the presence of immature cells with osteogenic capacity. Many of these PDGFRβ+ cells also express Runx2, but levels of the transcription factor vary considerably in the PDGFRβ+ population, indicating a spectrum of osteogenic differentiation (Supplementary Fig. 3g). Occasionally, in regions with dense fiber deposition, as visualized by SHG imaging, individual PDGFRβ+ Runx2+ cells show an extended, spindle-shaped morphology, suggesting migratory behavior (Supplementary Fig. 3g).

### Osteoblasts lining the bone front invade the vascularized calvarial lesion area

To visualize the dynamic behavior of osteoblast lineage cells in relation to ECs during bone healing, calvarial lesions in *Flk1-GFP, Sp7-mCherry* double reporter mice were repeatedly imaged by intravital multiphoton microscopy (Fig. 5a, Supplementary Fig. 5a). SHG signals indicate that early osteoblastic cells differentiate into bone-forming *Sp7-mCherry+* osteoblasts at the edge of growing bone, where they form a dense, multicellular layer lining the surface (Fig. 5a–c). *Sp7-mCherry+* osteoblastic cells are first visualized in the injured SHG+ calvarial bone at PLD6. *Flk1-GFP+* blood vessels are found close to the bone-lining *Sp7-mCherry+* osteoblastic cells and are part of the early vascular network that has penetrated the entire lesion area (Supplementary Fig. 5a). However, the *Flk1-GFP+* microvasculature in the uncalcified lesion area is not associated with *Sp7-mCherry+* osteoprogenitors (Fig. 5a, d, Supplementary Fig. 5a, b). Bone lining *Sp7-mCherry+* osteoblasts form new SHG+ bone matrix on pre-existing bone, as seen at PLD9, and often completely surround themselves with matrix fibers. At the expanding bone front, *Sp7-mCherry+* osteoblastic cells collectively invade the already vascularized lesion (Supplementary Fig. 5b, c). The leading front of invading *Sp7-mCherry+* osteoblastic cells is not surrounded by SHG+ fibers. However, the following *Sp7-mCherry+* osteoblasts initiate ossification and form a SHG+ fiber network leaving round and dark spaces containing their cell bodies (Supplementary Fig. 5b). In the following weeks, as seen at PLD14 and PLD21, ossification driven by *Sp7-mCherry+* osteoblasts progressively replaces the inner uncalcified lesion tissue with newly formed SHG+ bone tissue (Fig. 5a, Supplementary Fig. 5a). As ossification progresses, *Flk1-GFP+* microvessels are either gradually displaced or enclosed by the growing bone forming early BM cavities (Fig. 5a–c, Supplementary Fig. 5a, b). Cross sections at PLD14 confirm that a dense layer of *Sp7-mCherry+* osteoblastic cells lines the entire front of the growing SHG+ calvarial bone. In this region, *Sp7-mCherry+* osteoblasts are often completely surrounded by new SHG+ bone matrix. In the adjacent uninjured bone, *Sp7-mCherry+* osteocytes are less frequent and show lower levels of *Sp7-mCherry* expression (Fig. 5d–f).

Taken together, these results show that osteoblastic cells emerge from the peripheral edge of the flat bone injury and collectively migrate into the previously vascularized lesion to initiate ossification and gradually regenerate calvarial bone.

### Vascular and bone remodeling in calvarial lesions and femoral fractures

To better understand the role of Osx+ osteoblastic cells and their relationship to microvessels in two different bone types during bone regeneration, we compared calvarial bone lesions with long bone fractures at PLD14 and post-fracture day (PFD) 14, respectively (Fig. 6). The healing process in calvarial bone defects and femoral fractures shows clear differences with respect to the complexity of the regenerating tissue and the involvement of an intermediate avascular callus, which is only present in fractured long bones (Fig. 6a). A common feature of both types of injury is that Osx+ osteoblasts are positioned at the growing bone front, where they form new SHG+ bone matrix, often in proximity of microvessels (Fig. 6c–f). Lesion-associated vessels show high expression of CD31 (CD31high) and Emcn (Emcnhigh) in calvarial and femoral bone (Fig. 6b). To remodel and gradually replace the avascular callus in fractured femurs, growing vessels with a bud-shaped morphology, similar to the previously described CD31high Emcnhigh (type H) vessels at the growth plate, invade the intermediate cartilage at the chondro-osseous junction (Fig. 6a, c, d). Immediately behind this vascular growth front, Osx+ osteoprogenitors and osteoblasts are abundant in the perivascular area and deposit early collagen fibers, as visualized by SHG imaging (Fig. 6c). As endochondral ossification of the callus progresses, the ingrowing bone forms early SHG+ trabecular-like structures through continued fiber and matrix deposition followed by fiber fusion with adjacent collagen bundles to form larger trabecular entities. These early SHG+ collagen bundles are frequently intersected by vascular protrusions with a finger-like appearance (Fig. 6c, d). The healing process in calvarial lesions is clearly different from that of femoral fractures. Osx+ osteoblasts and progenitors in the calvarium line the anterior surface of the growing bone as a multicellular sheet and collectively invade the inner lesion area, which has been fully vascularized earlier (Fig. 6e, f). Bone-forming Osx+ osteoblasts continuously deposit new bone matrix as visualized by the dense SHG+ fiber network, while ossification proceeds inward to closure the lesion site (Fig. 6e, f). During this process, pre-existing *Flk1+* blood vessels are either physically displaced and remodeled or enclosed as early bone vessels (Figs. 5d, 6e, f).

Consistent with previous reports[50], PDGFRβ+ BM stromal cells in femoral fractures are closely associated with Emcn+ vessels at the osseous-chondral junction and are abundant in the early vascularized and calcified callus at PFD14 (Supplementary Fig. 4a). In calvarial bone lesions, PDGFRβ+ cells are highly abundant not only at PLD6 (Supplementary Fig. 3g), but also in uncalcified regions at PLD14. PDGFRβ+ cells frequently express Runx2, and have elongated nuclei (Supplementary Fig. 4b). Sp7+ osteoblastic cells lining the growing bone front and bone surfaces of small BM-like cavities do not express PDGFRβ. Thus, Sp7+ osteoblastic cells and PDGFRβ+ stromal cells occupy largely complementary regions within calvarial bone lesions (Supplementary Fig. 4c–e).

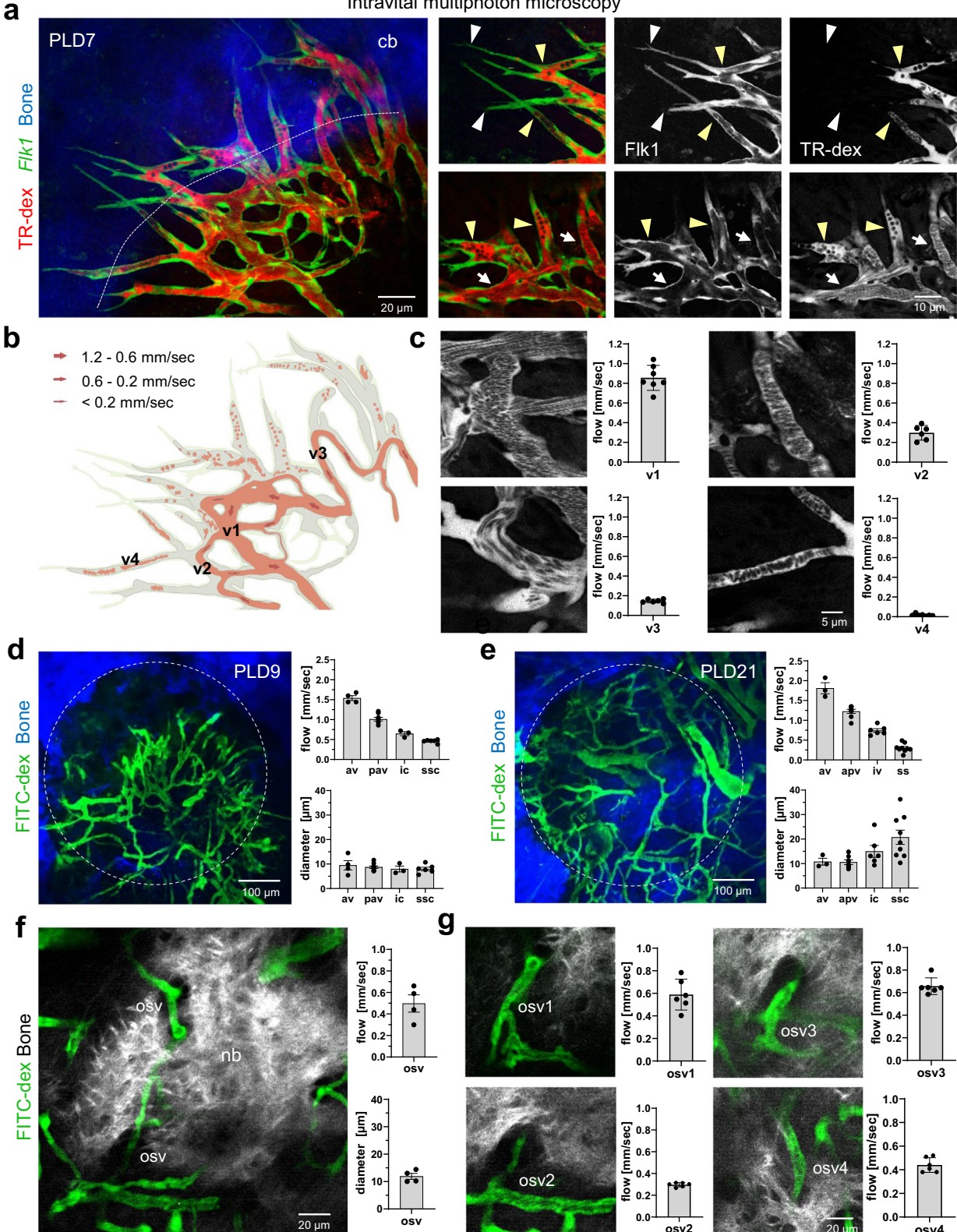

### Osteoclasts remodel regenerating bone in femoral fractures and calvarial bone lesions

Osteoclasts take on important functions in the complex process of bone regeneration after bone injury[51,52]. Vacuolar ATPases (vATPases), which are critical for extracellular acidification and thereby bone resorption, are a marker of mature osteoclasts. vATPase+ osteoclasts are positioned in fractured femurs at PFD14 in close proximity to bud-shaped vessels, which are invading the intermediate cartilage at the chondro-osseous junction (Fig. 7a). In addition, ATPase+ osteoclasts are abundant in the vicinity of early SHG+ bone matrix deposition and thus the remodeling vasculature of the hard callus (Fig. 7a). To address the role of osteoclasts in calvarial bone regeneration, we analyzed the locations of vATPase+ osteoclasts in calvarial bone lesions relative to Emcn+ microvessels and Sp7+ osteoblastic cells at PLD14 (Fig. 7b).

**Fig. 3 | Blood flow in sprouting and regenerating bone microvessels. a** Intravital multiphoton imaging of sprouting *Flk1-GFP*[+] (green) microvasculature near the SHG[+] (blue) calvarial bone edge in *Flk1-GFP*[+] reporter mice at PLD7. Intravenously injected TexasRed-dextran (red) visualizes the vascular lumen including circulating blood cells as dark round cells. White arrowheads indicate extended filopodia of *Flk1-GFP*[+] tip cells. Yellow arrowheads indicate luminal spaces of extended vascular sprouts. White arrows indicate perfused vessel segments. Data in **a-c** are from *n* = 3 biologically independent experiments. **b** Flow map of RBC velocities in sprouting microvasculature shown in **a** Flow velocities were grouped by flow rate. *Flk1-GFP*[+] endothelial lining is shown in light green. Perfused luminal spaces are shown in light red. Non-perfused luminal spaces (grey) are often found in vascular sprouts where flow is stationary. **c** Multiphoton images and histograms (right) of temporal variations of RBC flow velocities in a given vessel of the sprouting microvasculature shown in **a.** Vessel segments (v1-v4) are indicated in the flow map in **b** Data are presented as mean values ± SD. *n* = 7 over 3 biologically independent experiments.

**d, e** Blood flow velocities and vessel diameters in individual segments of the regenerating microvasculature at PLD9 (**d**) and PLD21 (**e**) according to previously defined flow velocities for different BM vessel types: arterial vessels (av), post-arterial vessels (pv), intermediate capillaries (ic) and sinusoidal capillaries (ssc)[41]. Data are presented as mean values ± SD, *n* = 3 (av), *n* = −8 (pav), *n* = 3 or 6 (ic), *n* = 6 or 9 (ssc) from 3 biologically independent experiments. **f** Multiphoton microscopy showing a SHG[+] growing bone edge at PLD21 with multiple GFP[+] microvessels in close proximity. Histograms with blood flow velocities and vessel diameters are shown. Data are presented as mean values ± SD. *n* = 4 over 3 biologically independent experiments **g** Multiphoton images and histograms (right) of temporal variation in RCB flow velocities in the indicated microvessels near the growing SHG[+] calvarial bone. Data are presented as mean values ± SD and show temporal variation in blood flow rate in the indicated vessel. *n* = 6 over 3 biologically independent experiments. Source data for Fig. 3c, d, e, f, g are provided as a Source Data file.

Multinucleated vATPase[+] osteoclasts are often found in small cavities of newly formed SHG[+] bone containing Emcn[+] microvessels, and are frequently located in close proximity to the bone surface. Some vATPase[+] osteoclasts are also found in close proximity to Emcn[+] microvessels. Notably, the distribution of Sp7[+] osteoblast clusters and vATPase[+] osteoclasts on the bone surface is mutually exclusive (Fig. 7b, e). MMP9[+] staining is abundant in these early bone cavities and is associated with multinucleated cells with osteoclast morphology, suggesting bone resorptive activities (Fig. 7c). Osteoclastic activity extends early BM-like cavities into the adjacent uninjured calvarial bone bridging the gap between 'old' and newly formed bone (Fig. 7b). vATPase[+] osteoclasts are also found in BM cavities of uninjured bone, where they often occupy opposite surfaces within the BM cavity[39], indicating an ongoing remodeling process (Fig. 7d). Our data suggest that osteoclasts remodel early calvarial bone and vascularized cavities in regenerating bone.

Thus, regeneration of calvarial and femoral bone injuries both involves bone-resorbing and remodeling osteoclasts and bone-forming osteoblasts, often in close proximity to microvessels.

## Single-cell profiling of bone non-hematopoietic cells after calvarial bone injury

To gain insight into the cellular and molecular changes in non-hematopoietic cells after calvarial bone injury at PLD14, we performed scRNA-seq analysis of lesioned areas and compared the resulting transcriptional profiles with uninjured and age-matched controls (Fig. 8a). Cluster-specific marker genes characterize different cell populations (Fig. 8b–d), including bone mesenchymal stromal cells (BMSCs), osteoblastic cells (OBs), fibroblasts (FBs), smooth muscle cells (SMCs), and endothelial cells (ECs). The number of cells in the BMSC and OB cluster is substantially increased in PLD14 calvarium (Fig. 8e, f), reflecting the ongoing healing process. Transcripts for platelet-derived growth factor receptor β (*Pdgfrb*), Runt-related transcription factor 2 (*Runx2*), Osterix (*Sp7*), and type XXII collagen (*Col22a1*), which is a recently identified regulator of osteogenesis[53] are all upregulated in PLD14 BMSCs and osteoblast lineage cells relative to control (Fig. 8g). Two fibroblast subpopulations can be distinguished (Fig. 8e, f, Supplementary Fig. 6a, b), one of which (FBs-1) remains largely unchanged in number during regeneration and expresses markers of resident fibroblasts (*Thy1, Col3a*) (Fig. 8g)[54]. The second fibroblast population (FBs-2) shows expression of insulin-like growth factor-binding protein 6 (*Igfbp6*), is rare in uninjured samples and profoundly enriched after lesion injury, pointing to a potential role in bone healing and tissue remodeling (Fig. 8e–g). Fibroblasts commonly express high levels of collagens, including collagen I (*Col1a1*) and collagen III (*Col3a1*), and show at PLD14 upregulation of collagen VIII (*Col8a1*), a non-fibrillar lattice-forming collagen that is highly upregulated in vascular injury[55]. Upregulation is also seen for fibronectin (*Fn1*), a major fibrillar ECM glycoprotein that binds a variety of growth

factors relevant to bone regeneration, including VEGF-A (*Vegfa*) (Supplementary Fig. 6c)[56].

The number of ECs is also higher in PLD14 samples, reflecting increases in multiple EC subclusters relative to control. This includes tip cell ECs (tECs), arterial ECs (aECs), bone marrow (bmECs), skull ECs (skECs, comparable to metaphyseal mpECs in femur[50]) and osteogenic ECs (osECs) (Supplementary Fig. 7a–d). During calvarial bone repair, osECs show strong expression of genes that support vascular growth and bone regeneration, i.e. Basigin/ EMMPRIN (*Bsg*), a potent inducer of VEGF signaling[57], and osteonectin (*spock2*), a bone-specific glycoprotein that selectively binds to both hydroxyapatite and collagen with an active role in bone tissue mineralization (Supplementary Fig. 7e)[58]. A subcluster of arterial-like ECs (alECs-1), a population that has been also observed in long bone[50], is largely absent in PLD14 samples (Supplementary Fig. 7a). *Pecam1* and *Emcn* expression are moderately upregulated at PLD14 (Supplementary Fig. 7f). *Pecam1* is highly expressed in arterial and arterial-like ECs, whereas *Emcn* expression is preferentially found in bmECs, skECs and tECs, but not in arterial aECs (Supplementary Fig. 7f), as previously shown[31].

To examine how hypoxia and vascularization influence gene expression in healing calvarial bone, we analyzed *Hif1a, Vegfa* and *Angpt2* expression in various cell populations at PLD14. While *Hif1a* and *Angpt2* expression are essentially unchanged, *Vegfa* expression is slightly increased in MSCs and osteoblasts (OBs) in the injured calvarium compared to controls (Supplementary Fig. 6d, e), reflecting that the lesion is completely vascularized at PLD14. In controls, *Vegfa* shows a higher expression in bmMSCs compared to other stromal cells suggesting a possible role in vascular remodeling in the uninjured bone. *Hif1a* expression is generally low in calvarial BMSCs at PLD14 and controls (Supplementary Fig. 6d, e).

Next, we compared *Hif1a, Vegfa* and *Angpt2* expression in different cell populations within calvarial lesions with femoral fractures (Supplementary Fig. 8a–d). The two bone injury models show clear differences in respect to vascularization and *Hif1a* expression. While *Hif1a* is not increased in calvarial lesions at PLD14 (Supplementary Figs. 6e, 8e), the transcript is strongly upregulated in BMSC populations from femoral fractures (Supplementary Fig. 8e, f), reflecting the presence of avascular cartilage and ongoing endochondral ossification (Fig. 6a).

Accordingly, *Vegfa* expression is increased in chondrocytes (Supplementary Fig. 8f), particularly in hypertrophic chondrocytes (Supplementary Fig. 8g, h) that are located in proximity to vascular protrusions at the chondro-osseous junction (Fig. 6a)[22].

We provide access to the scRNA-seq data on calvarial lesions through an online cell viewer platform under the following link https://single-cell.mpi-muenster.mpg.de/o/calvarial-lesion-2023. This allows further exploration of the changes in cellular composition and gene expression resulting from the regenerative ossification in calvarial lesions at PLD14.

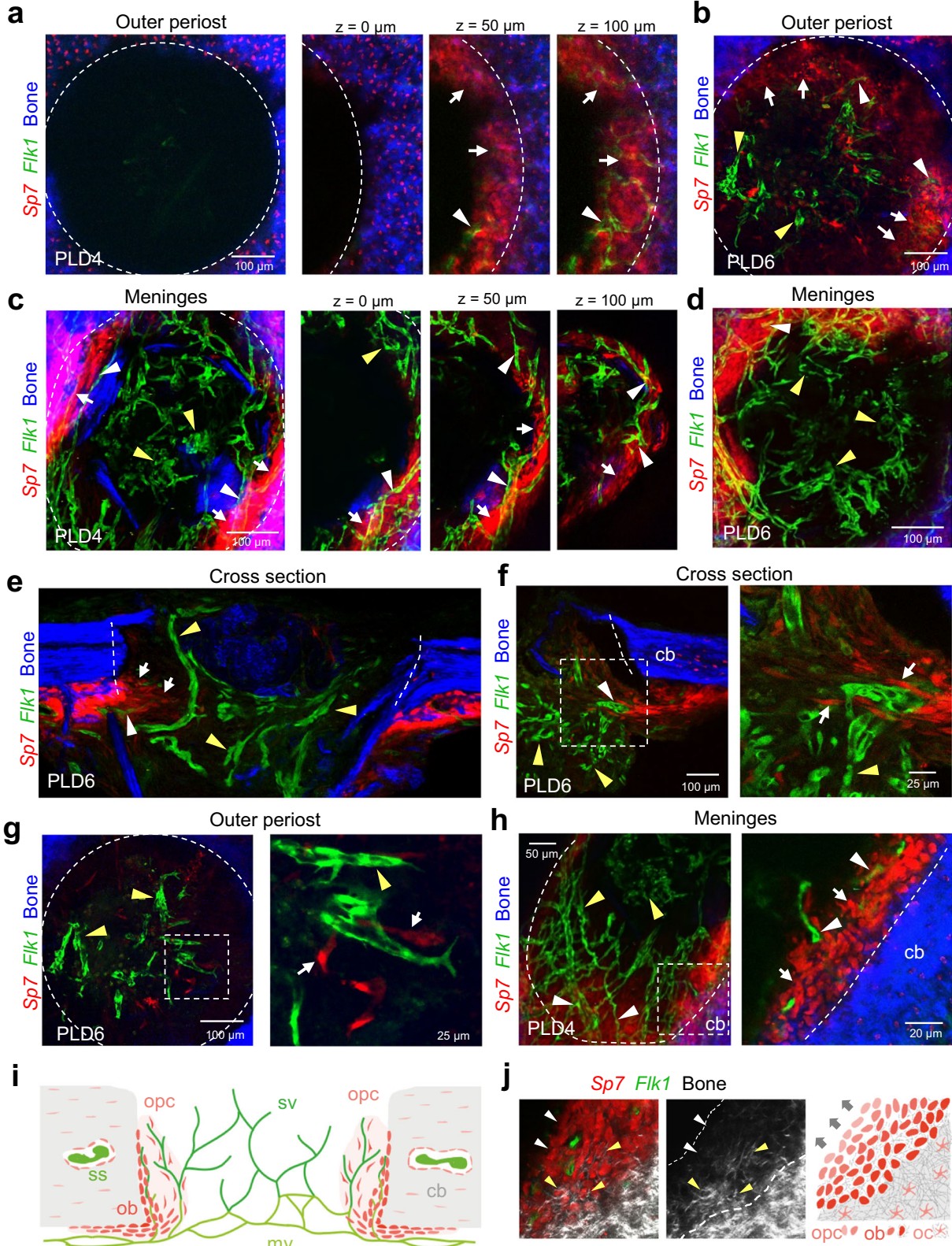

**Notch and Vegfa signaling alter vascular regeneration without affecting calvarial bone healing**

Notch signaling is a negative regulator of angiogenic vessel growth in most organs, but in the long bone, the pathway promotes the expansion of CD31$^{high}$ Emcn$^{high}$ (type H) capillaries and couples angiogenesis to osteogenesis through the secretion of EC-derived paracrine (angiocrine) factors[17,36]. To investigate the possible influence of endothelial Notch signaling on angiogenesis and bone healing in the calvarium, we inactivated the gene encoding F-Box and WD Repeat Domain Containing 7 (FBXW7), which mediates the proteasomal degradation of active Notch and thereby limiting its signaling activity[59]. Adult EC-specific mutants (*Fbxw7*$^{iΔEC}$), generated by interbreeding of mice carrying floxed *Fbxw7* alleles[60] and the *Cdh5-CreERT2* transgenic line[61], were analyzed in the *Flk1-GFP* reporter background

**Fig. 4 | Osteoprogenitors emerge from the periosteum and collectively colonize the calvarial bone lesion, while blood vessel vascularizes the entire lesion.** **a, b** Multiphoton imaging of wholemount calvarial bone at PLD4 (**a**) and PLD6 (**b**) from the outer periosteal side. **a** *Sp7-mCherry*⁺ (red) osteoblastic cells colonize the injured SHG⁺ (blue) calvarial bone. Osteoprogenitors are not found at the outer bone edge at PLD4 (z = 0 μm). Deeper layers (≥ 50 μm) show early bone-lining osteoprogenitors (white arrows) and associated *Flk1*⁺-*GFP* microvessels (white arrowheads). **b** The majority of *Sp7-mCherry*⁺ osteoprogenitors line the injured calvarial bone (white arrows). Most *Flk1*-*GFP* microvessels in the lesion center are not associated with osteoprogenitors (yellow arrowheads). **c, d** Multiphoton imaging of wholemount calvarial bone at PLD4 (**c**) and PLD6 (**d**) from the inner meningeal side. Early *Sp7-mCherry*⁺ osteoprogenitors (white arrows) line the injured bone close to blood vessels (white arrowheads). Microvessels in the lesion center are not associated with osteoprogenitors (yellow arrowheads). **e–g** Cross sections (**e, f**) and intravital micrograph (**g**) of calvarial bone at PLD6 showing early *Flk1-GFP*⁺ microvessels and invading, spindle-shaped *Sp7-mCherry*⁺ osteoprogenitors. Osteoprogenitors (white arrows) originating from the periosteal layer collectively invade the lesion close to the calvarial bone. Note microvessels with associated osteoprogenitors (white arrowheads), while microvessels in the lesion center are frequently solitary (yellow arrowheads). **h** Multiphoton imaging showing the meningeal calvarial bone side at PLD4. *Flk1*⁺-*GFP* microvessels (white arrowheads) near the injured calvarial bone are surrounded by *Sp7-mCherry*⁺ osteoprogenitors (white arrows). In the lesion center, vessels are not associated with osteoprogenitors (yellow arrowheads). **i** Schematic showing calvarial bone lesion at PLD6 with sprouting vessels (sv) originating from meningeal vessels (mv) and invading osteoprogenitor (opc). Osteoprogenitors remain close to the injured calvarial bone, while sprouting vessels penetrate and vascularize the entire lesion. Area of angiogenic-osteogenic coupling is shown in light red. Microvessels in the center of the lesion are devoid of osteoprogenitors. ss: sinusoidal vessels, ob: osteoblasts, oc: osteocytes. **j** Multiphoton imaging showing the growing calvarial bone edge and corresponding schematic. *Sp7-mCherry*⁺ osteoprogenitors invade the bone lesion as a multicell layer and gradually differentiate to matrix forming osteoblasts. Grey arrow indicates direction of bone growth. Reproducibility was ensured by *n* = 3 or more biologically independent experiments.

by longitudinal intravital multiphoton imaging after drill hole injury (Fig. 9a, Supplementary Fig. 9a). Around PLD6, the first *Fbxw7*^iΔEC sprouts appear in the bone lesion and sprouting angiogenesis rapidly expands the vascular network (Fig. 9a–d, Supplementary Fig. 9b). The vascular area of *Fbxw7*^iΔEC mutants is greatly increased compared to controls (Fig. 9c), demonstrating that angiogenesis in flat bone is promoted by endothelial Notch signaling. However, the majority of the newly formed mutant vessels are highly elongated at PLD9, lack a lumen and do not carry blood, whereas microvessels in controls are luminized (Fig. 9a–c, Supplementary Fig. 9c). The *Fbxw7*^iΔEC mutant vasculature inside lesions shows a 2-3-fold increase in vascular sprouts (Fig. 9d). Unlike vascular sprouts in controls, *Fbxw7*^iΔEC tip cells are atypically thickened and extend blebbing protrusions at their surface (Supplementary Fig. 9d). Gradually lumen formation in *Fbxw7*^iΔEC normalizes and hypersprouting decreases, reaching control levels by 4-5 weeks post-lesion (Fig. 9d).

After calvarial bone injury, the onset and progression of ossification in *Fbxw7*^iΔEC mutants is largely comparable to controls with a slight delay at later stages (Fig. 9e, f). In both, mutants and controls, newly formed SHG⁺ bone matrix forms an interconnected fiber network and surrounds early osteocytes, which are visualized as round and dark speckles in the SHG⁺ bone matrix (Fig. 9e). As a result of the vascular hypersprouting, numerous *Fbxw7*^iΔEC microvessels are observed in close proximity to the growing bone, but the majority is not perfused (Fig. 9e).

We performed scRNA-seq analysis to investigate whether hypersprouting in *Fbxw7*^iΔEC mutants affects gene transcripts related to hypoxia and ECM formation. Both, *Hif1a* and *Vegfa* expression in *Fbxw7*^iΔEC mutants are comparable to controls at PLD14 (Supplementary Fig. 10a, b). The expression of collagens, including the major fibrillar collagens *Col1a1* and *Col3a1* and the vascular injury related collagen *Col8a1*, is basically unchanged. Furthermore, ECM components involved in bone regeneration, i.e. *Col11a1*, *Fn1* and *Bsg* are either unchanged or slightly increased in *Fbxw7*^iΔEC mutants compared to controls (Supplementary Fig. 10c). Similarly, transcripts of MSC and OB differentiation markers are comparable between *Fbxw7*^iΔEC mutants and controls (Supplementary Fig. 10d), suggesting that MSC differentiation into osteoblasts is not affected. Thus, EC-specific Notch activation results in vascular hypersprouting and delayed lumen formation, but has no beneficial effect on ossification in calvarial lesions.

To further investigate the relationship between angiogenesis and osteogenesis during calvarial bone healing, we inactivated the expression of the Notch ligand Delta-like 4 (Dll4) in endothelial cells to inhibit Dll4-Notch interactions[62]. Adult EC-specific mutants (*Dll4*^iΔEC), generated by interbreeding of mice carrying floxed *Dll4* alleles and *Cdh5-CreERT2* transgenic mice, were analyzed after drill hole injury at PLD14. Inhibition of endothelial Notch signaling resulted in an increase in vascular density by 1.9-fold, while the progression of bone healing was unaffected (Fig. 10a, b). The regenerating *Dll4*^iΔEC vasculature at PLD14 shows hyperbranching and vessel maturation defects as previously reported for other vascular beds[63].

In addition, we overexpressed VEGF-A, a key vascular growth factor, during lesion experiments to promote vascular growth and investigate its effect on calvarial bone healing. To this end, a cDNA encoding a bone-homing version of *Vegfa* was cloned into the pLIVE vector to allow constitutive protein expression in the liver after hydrodynamic tail vein injection[64]. *Vegfa* overexpression results in a 1.6-fold increase in vascular area at PLD14, with some microvessels showing a partially compromised lumen (Fig. 10c, d). Similar to the findings in Notch pathway mutants, the progression of bone healing was not affected by systemic overexpression of VEGF-A (Fig. 10d). In conclusion, endothelial Notch and Vegfa signaling during calvarial bone healing affects vascular sprouting and remodeling without affecting the ossification process.

## Discussion

To visualize complex biological events and directly observe cellular dynamics in the bone and BM environment in living mice, various intravital imaging approaches have been developed over the last decade[35,37,41,42]. Intravital multiphoton imaging of bone regeneration is highly challenging due to the optically dense bone tissue and the slow time course of bone regenerative processes. To date, longitudinal intravital imaging using high-resolution multiphoton microcopy of calvarial bone repair has not been reported at a multiscale level in the same animal until bone healing is complete. Our advanced imaging approach captures the early and highly dynamic sprouting events related to regenerative ossification in calvarial bone lesions. Critical to this approach is the minimal invasiveness of the optical window combined with the high spatial resolution of deep tissue multiphoton microscopy and the repetitive imaging over a prolonged period of time. Alternative methods using skin flaps induce significant inflammatory responses and disrupt the tissue microarchitecture of the regenerating bone due to repetitive surgical procedures prior to each imaging session[65]. Previous work has shown that the entry of osteoblast precursors in developing and regenerative osteogenesis in long bone is associated with the invasion of growing microvessels[49]. Subsequent work has highlighted vascular heterogeneity in long bone by showing that specialized CD31^hi Emcn^hi capillaries promote osteogenesis through interactions with perivascular osteoprogenitor cells[14]. A similar link between angiogenesis and osteogenesis can be observed in fractured bone where osteoblast precursors are located in close proximity to newly formed blood vessels[13,14]. VEGF is required for blood vessel invasion into the hypertrophic cartilage of the fracture callus and promotes the replacement of cartilaginous structures by a

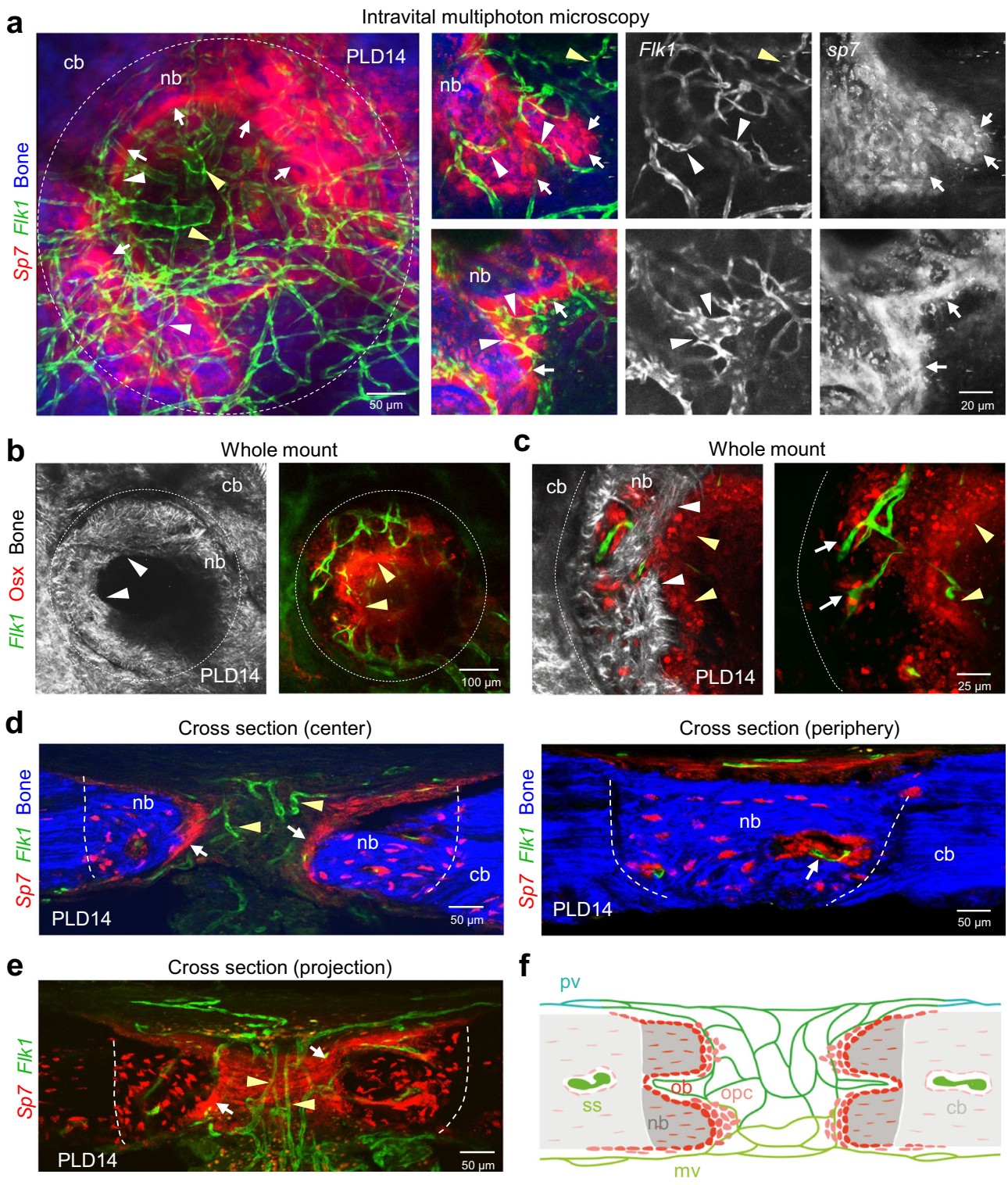

**a** Intravital multiphoton microscopy

**b** Whole mount

**c** Whole mount

**d** Cross section (center) / Cross section (periphery)

**e** Cross section (projection)

**f**

bony callus[16,66,67]. In calvarial lesions, a recent report showed that regeneration involves rapid expansion of CD31⁺ Emcn⁺ microvessels, which is followed by their maturation into BM vessels[31]. The same study also proposed that EC coupled to osteoblasts prefer glycolysis, whereas osteoblasts rely more on oxidative phosphorylation and potentially consume more oxygen[31]. Our own data confirm that early vascular sprouting coincides with osteoprogenitor expansion, but we show that the collective invasion of osteoprogenitors and osteogenesis are not directly coupled to endothelial sprouting and occur at different time points during calvarial bone healing. Thus, the

regenerative processes and the role of angiogenic vessel growth are substantially different in flat bone and long bone or during intramembranous and endochondral ossification, respectively.

Early vascularization of the calvarial bone lesion strongly influences the gene expression profile of oxygen-related genes in bone stromal cells during the healing process. Our scRNA-seq data show that hypoxia-related genes, i.e. *Hif1a*, *Vegfa* and *Angpt2* are basically unchanged or slightly upregulated at PLD14, reflecting that calvarial ossification and tissue remodeling proceeds into a fully vascularized and perfused lesion, which is very different from femoral fractures.

**Fig. 5 | Osteoblasts lining the growing bone collectively invade the vascularized calvarial bone lesion. a** Intravital multiphoton microscopy showing the expanding SHG⁺ (blue) calvarial bone lined by a multicellular layer of *Sp7-mCherry*⁺ (red) osteoblastic cells and *Flk1-GFP*⁺ (green) microvessels vascularizing the calvarial bone lesion in *Flk1-GFP, Sp7-mCherry* double transgenic mice at PLD14. Maximum intensity projection (left). Zoom-in views (right) show association of microvessels (white arrowheads) with a dense layer of osteoblastic cells at the front of the growing bone (nb) (white arrows). Note that microvessels in the uncalcified lesion center are not associated with osteoprogenitors (yellow arrowheads). **b, c** Multiphoton microscopy of whole mount calvarial bone at PLD14. **b** Maximum projection shows Osx⁺ (red) osteoblasts and progenitors, *Flk1-GFP*⁺ (green) microvessels, SHG⁺ (white) calvarial bone (cb) and new bone (nb) in *Flk1-GFP*⁺ transgenic mice stained with anti-Osx antibodies. **c** Zoom-in views show single planes. White arrowheads point to the front of growing SHG⁺ bone matrix (nb). Yellow arrowheads indicate the front of a multicellular layer of Osx⁺ osteoblastic cells that precedes the SHG⁺ bone front. White arrows point to microvessels enclosed in the newly formed bone in close proximity to Osx⁺ osteoblastic cells. **d** Cross sections

showing *Flk1-GFP*⁺ (green) microvasculature, *Sp7-mCherry*⁺ (red) osteoblasts and progenitors lining the SHG⁺ (blue) growing bone, osteocytes enclosed by SHG⁺ new bone (nb) at PLD14. Two single planes (left: center, right: periphery) through the lesion are shown. White arrows (right) indicate the multicellular layer of osteoblasts and progenitors lining the new bone. Yellow arrowheads indicate microvessels. Arrow (left) indicates an early bone marrow cavity. **e** Maximum intensity projection of a cross section showing *Flk1-GFP*⁺ (green) microvasculature, *Sp7-mCherry*⁺ (red) osteoblasts and progenitors at PLD14. White arrows indicate osteoblasts at the front of the growing bone. Yellow arrows indicate microvessels connecting meningeal and periosteal vessels in the uncalcified lesion. **f** Schematic cross section of calvarial bone at PLD14. Meningeal vessels (mv) form a vascular network within the lesion that connects to outer periosteal vessels (pv). Osteoblastic cells expand the growing bone into the vascularized lesion area. Microvessels in the uncalcified lesion are devoid of osteoprogenitors. cb: calvarial bone, oc: osteocytes ob: osteoblasts, opc: osteoprogenitors, ss: sinusoidal capillaries. Reproducibility was ensured by *n* = 3 or more biologically independent experiments.

During endochondral ossification, recruitment of mesenchymal progenitor cells that differentiate into chondroblasts contributes to avascular callus formation[22]. Hypertrophic and hypoxic chondrocytes induce *Vegfa*-dependent blood vessel invasion into avascular cartilage to allow callus remodeling into early bone tissue[66,68]. Accordingly, our scRNA-seq data confirm that *Hif1a* is substantially upregulated in bone stromal cells and *Vegfa* is increased in CHOs, particularly in hypertrophic CHOs at PFD14.

The structure of the extracellular matrix (ECM) plays an important role in tissue regeneration, including bone healing after injury, and there is clear evidence that extracellular structures are coupled to cell migration and differentiation[69,70]. A recent report suggests that collagen fiber orientation has a pre-patterning function, guiding tissue mineralization after osteotomy during regeneration of large bone defects[44]. The authors describe early patterns of collagen fibrils in periosteal and endosteal regions that tend to cross the marrow space. Similarly, we observe the early appearance of a thin collagen network that penetrates the entire lesion prior to calvarial bone ossification, possibly formed by early fibroblasts based on their gene expression of collagens (*Col1a1, Col3a1, Col8a1*) and fibronectin (*Fn1*). During tissue mineralization and new bone matrix formation, these thin and parallel collagen fibers are replaced by a differently organized collagen network formed by bone-forming osteoblasts, consisting mainly of collagen type I (Col1a1). Collagen type III (Col3) has been proposed to play a key role in tissue repair based on its expression pattern during healing processes, including bone regeneration[71]. Mice with reduced Col3 expression show an increased callus volume after long bone fracture, but with less bone, suggesting a role for Col3 in osteogenesis[72]. In calvarial lesions, *Col3a1* gene expression is increased in the FB-2 and MCS populations, but it remains to be shown whether Col3 has a functional role in calvarial bone healing. Osteoclasts are critical for the resorption of calcified bone and play a significant role in bone remodeling during fracture healing[51,52]. Consistent with previous reports[73], we confirm the abundant appearance of osteoclasts in the periosteal callus of fractured femurs suggesting their active participation in the healing process. In calvarial bone repair, the contribution of osteoclasts after bone injury is less well understood. Our data show that osteoclasts play an active role in the early phase of calvarial bone healing when growing bone surrounds remodeling microvessels in the lesion and forms early BM-like cavities. Osteolytic activity by osteoclasts may provide a mechanism by which the gap between 'old' and new bone could heal by forming new cavities that bridge both sides, thereby creating space for osteoblastic activity and new bone formation.

Mechanobiological regulation in ECs has been extensively studied, demonstrating that blood flow provides mechanical cues that drive vascular remodeling[74] and vessel regression[75]. After the calvarial bone injury, we observe robust revascularization and restoration of

blood flow in the expanding vascular network. The highly variable blood flow dynamics observed in the early vasculature likely contributes to vascular remodeling and gradual functional differentiation of BM microvessels. In addition, the growing bone physically displaces and thereby remodels microvessels in the immediate vicinity, or alternatively surrounds them with bone matrix. A recent study attempted to correlate local oxygen tension in regenerating calvarial bone lesions with the caliber of nearby microvessels and found that vessel size and microvascular density are poor indicators of local oxygen tension[31]. Our own data show that RCB perfusion varies considerably between individual microvessels, contributing to highly variable blood flow dynamics in the regenerating BM compartment.

Endothelial Notch activity is critical for promoting angiogenesis and osteogenesis in developing bone[17]. Additionally, VEGF and hypoxia signaling components have been described as important players in angiogenic-osteogenic coupling[13,16]. However, at the molecular level, the precise mechanism of angiogenesis and its coupling with osteogenesis during fracture repair is not completely understood[15]. In contrast to fractured femurs, we observe in three different models during calvarial bone repair, i.e. Notch activation and inactivation, and *Vegfa* overexpression, that profound vascular changes have little effect on bone regeneration, possibly due to uncoupling of endothelial sprouting from collective invasion of osteoprogenitors. A more detailed analysis of this striking difference between calvarial and femoral bone healing is required to understand the molecular mechanism behind the uncoupling of vascular sprouting from co-migrating osteoprogenitors. A deeper understanding of the highly complex regenerative processes after bone injury is critical for the long-term success of strategies to improve bone regeneration and overcome major challenges in orthopedic surgery. Here, we establish a unique longitudinal intravital multiphoton imaging approach to visualize calvarial bone regeneration in living mice, including dynamic sprouting, remodeling and maturation of the calvarial bone vasculature and gradual new bone formation through the collective activity of osteoblastic cells. We show that sprouting blood vessels form a vascular network in the lesion before the entry of osteoprogenitors, implying that angiogenesis and osteogenesis occur at different stages of the repair process. Sprouting vessels originating from the meningeal layer vascularize the entire lesion, while osteoprogenitors gradually mediate ossification from the edge of the injured bone. Thus, the process of calvarial bone regeneration clearly differs from long bone fracture healing with respect to the functional role of angiogenic vessels, the co-invasion of vessels and osteoprogenitors, and the expression of oxygen-related genes. Our comprehensive study of vascular regeneration during calvarial bone healing provides fundamental insight into the regeneration of flat bone, which has important implications for bioengineering but also for the development of future therapeutic strategies to improve bone regeneration.

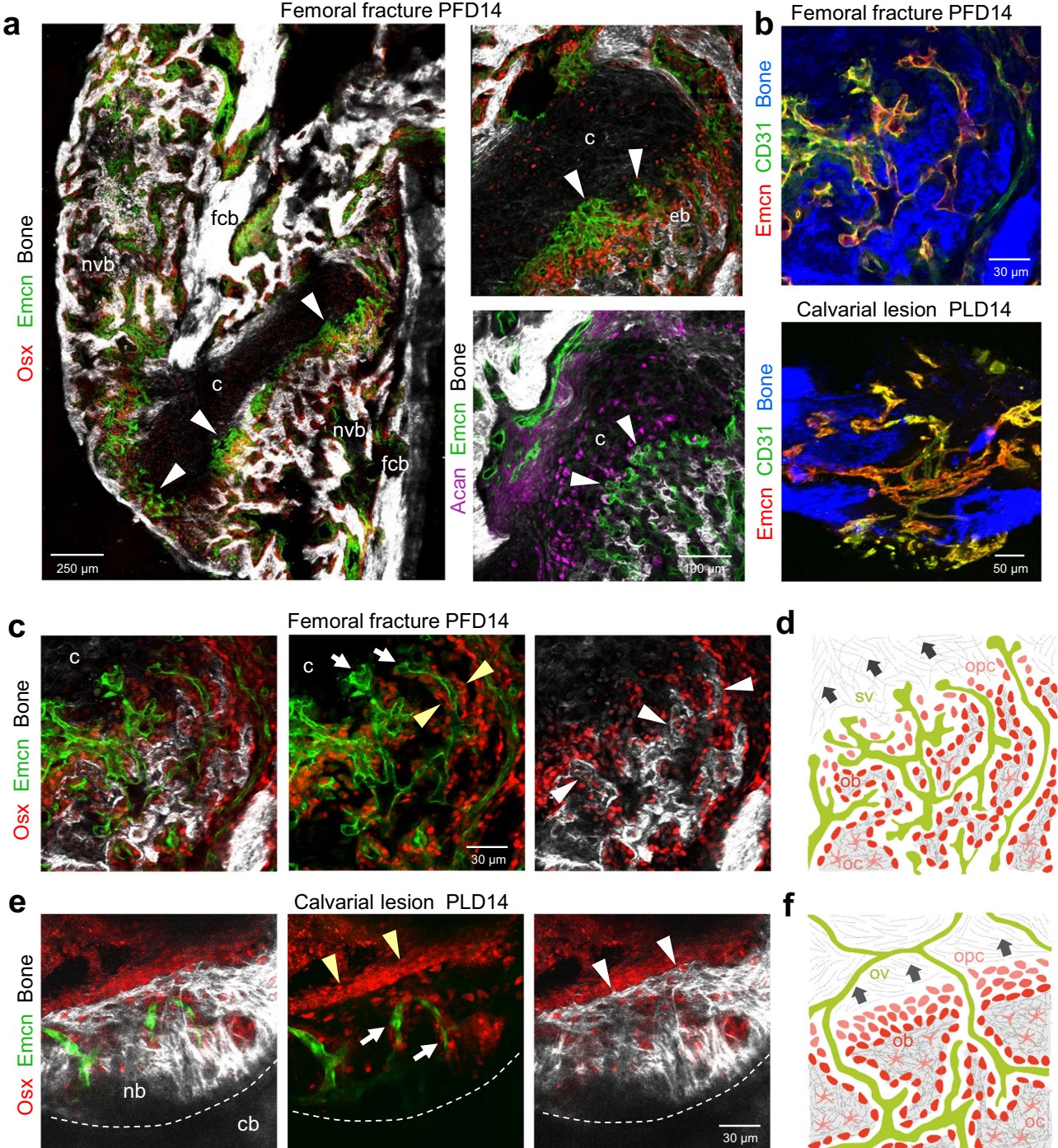

## Methods

### All animal experiments were performed according to the institutional

All animal experiments were performed according to the institutional guidelines and laws, approved by local animal ethical committee and were conducted at the University of Münster and the Max Planck Institute for Molecular Biomedicine with necessary permissions (84-02.04.2018.A171, 81-02.04.2019.A164) granted by the Landesamt für Natur, Umwelt und Verbraucherschutz (LANUV) of North Rhine-Westphalia, Germany.

### Multiphoton imaging setup

We used a TriM Scope II multi photon system from LaVision BioTec (Bielefeld, Germany) to visualize immune labeling and SHG generated by collagen in the bone[41]. The setup is a single-beam instrument with an upright Olympus BX51 WI microscope stand that is equipped with highly sensitive NDD detectors close to the objective lens. The TriM Scope II is fitted with a Coherent Scientific Chameleon Ultra II Ti:Sapphire laser and a Coherent Chameleon Compact OPO. A 20x IR objective lens (Olympus XLUMPlanFL; NA 1.0) with a working distance of 2.0 mm was used. The microscope is equipped with a pair of x-y galvanometric mirrors used to scan at a scanning speed of up to 1200 lines/s. The maximal laser power (850 nm) on the object was 10–20 mW at superficial areas and 80–100 mW when imaging at large depths. Dichromatic mirrors and band pass filters spectrally separate the emitted light before the signal is detected using attached photomultiplier tubes (PMT; Hamamatsu H67080-01 (blue channel), H67080-20 (green and red channels). The following detection channels were used: red (560–680 nm), green (475–575 nm) and blue (370–470 nm). 3D-images were acquired and processed with LaVision BioTec ImSpector Pro.

**Fig. 6 | Vascular and bone regeneration in femoral fractures and calvarial bone lesions. a** Tile-scan multiphoton microscopy of a PFD14 femoral fracture. Overview image shows maximum intensity projections of Emcn⁺ (green) microvasculature, Osx⁺ (red) osteoblasts and progenitors and SHG⁺ (white) newly formed trabecular-like woven bone (ntb) and fractured compact bone (fcb). Zoom-in views (right) show early Emcn⁺ microvessels (arrowheads) invading the Acan⁺ (purple, aggrecan) cartilage (c) in close proximity to Osx⁺ osteoprogenitors. Early SHG⁺ bone matrix depositions (eb) at the chondro-osseous junction. White arrowheads indicate bud-shaped vascular sprouts. **b** Multiphoton microscopy showing Emcn⁺ (red) and CD31⁺ (green) microvessels at the chondro-osseous junction of a femoral fracture at PFD14 (top) and of a calvarial lesion at PLD14 (bottom). **c** Multiphoton microscopy of femoral fracture at PFD14 showing Emcn⁺ microvessels in adjacent Osx⁺ osteoprogenitors. White arrows point to bud-shaped microvessels invading cartilaginous tissue (c). Early finger-shaped SHG⁺ bone matrix deposits (white arrowheads). Yellow arrowheads point to osteoblasts adjacent to early bone matrix deposits. **d** Schematic representation of the process of endochondral ossification after femoral fracture with sprouting vessels (sv) invading the hypertrophic cartilage at the chondro-osseous junction. Osteoprogenitors (opc) co-migrate in close proximity to the invading vessels. Osteoprogenitors differentiate into early osteoblasts (ob) that deposit collagen fibers and trabecular-like bone matrix. Osteoblast that are enclosed by bone matrix differentiate to osteocytes (oc). Dark arrows indicate the direction of the ossification process. **e** Multiphoton microscopy of PLD14 calvarial bone lesion showing Emcn⁺ microvessels, multicellular layer of Osx⁺ osteoblasts and progenitors leading the SHG⁺ growing bone edge. White arrowheads point to SHG⁺ growing bone edge. Yellow arrowheads point to multicellular layer of Osx⁺ osteoblasts and progenitors. **f** Schematic representation of the process of intramembranous ossification after calvarial lesion injury with osteoblastic cells collectively invading the vascularized lesion tissue as a multicellular layer. Osteoprogenitors (opc) differentiate into osteoblasts (ob) and form new bone matrix. Blood vessels in the lesion are either remodeled or surrounded by the growing bone. Specialized osteogenic vessels (ov) are located close to the growing bone front. Dark arrows indicate the direction of the ossification process. Reproducibility was ensured by $n = 3$ or more biologically independent experiments.

## Fluorescence microscopy

Bone immunostaining performed as described previously[76], bone sections were washed in ice-cold PBS and permeabilized with ice-cold 0.3% Triton-X-100 in PBS for 10 mins at room temperature (RT). Samples were incubated in blocking solution (5% heat-inactivated donkey serum in 0.3% Triton-X-100) for 1 h at RT. Primary antibodies: rat monoclonal anti-Endomucin (1:200, Santa Cruz, Cat# sc-65495, clone V.7C7), goat polyclonal anti-CD31 (1:200, R&D, Cat# AF3628), goat polyclonal anti-Pdgfrb (1:200, R&D, Cat# AF1042), rabbit poly-clonal anti-Osterix (1:200, Abcam, Cat# ab22552), rabbit polyclonal anti-Acan (1:200, Millipore, Cat#AB1031), rabbit polyclonal anti-vAT-PaseB1/B2 (1:200, Abcam, Cat# 200839), rabbit polyclonal anti-Runx2 (1:200, Abcam, Cat# 192256) were diluted in 5% donkey serum mixed PBS and incubated overnight at 4 °C. Next, slides were washed 3 to 5 times in PBS in 5-10 min intervals. Wholemount calvarial bones with drill hole lesions were fixed with PFA (2%) overnight at 4 °C and stained using primary antibody rabbit polyclonal anti-Osterix (1:200, Abcam, Cat# ab22552) for 3 days at 4 °C to allow for sufficient tissue penetration of the antibodies. Species-specific Alexa Fluor secondary antibodies (goat anti-rat IgG Alexa Fluor 594 (1:200, Thermo Fischer Scientific, Cat# A21209), donkey anti-rabbit IgG Alexa Fluor 647 (1:200, Thermo Fischer Scientific, Cat# A31573), donkey anti-goat IgG Alexa Fluor 647 (1:200, Thermo Fischer Scientific, Cat# A21447), donkey anti-rat IgG Alexa Fluor 488 (1:200, Thermo Fischer Scientific, Cat# A21208) were diluted in PBS were added and incubated for 3 h at RT or overnight at 4 °C.

## Construction of fluorescent composite and tile scan images

Multiphoton images obtained with the microscope ImspectorPro acquisition software were analyzed, processed and assembled using ImageJ (open source NIH software, http://imagej.nih.gov/ij). For tile scans images a 20% overlap were recorded and stitched using the stitching plugin from ImageJ.

## Animal models

All mice used were on a *C57/Bl6J* background. Female mice were used between 8–12 weeks old unless stated otherwise. *Flk1-GFP* ⁺ transgenic mice (STOCK Kdrtm2.1Jrt/J, RRID:IMSR_JAX:017006)[77] were used to label the bone vasculature. *Sp7-mCherry*⁺ transgenic mice (STOCK Tg(Sp7/mCherry)2Pmay/J, RRID:IMSR_JAX:024850)[78] were used to visualize osteoprogenitor and osteoblasts. Vascular reporter *Flk1-GFP* mice were crossed to Fbxw7 lox/lox:Cdh5Cre-ERT2 (Fbxw7^iΔEC) (B6;129-Fbxw7tm1Iaai/J, RRID:IMSR_JAX:017563: Tg(Cdh5-cre/ERT2) 1Rha, MGI:3848982)[79,80] to study Notch GOF in endothelial cells. Notch LOF in endothelial cells was studied in Dll4 lox/lox:Cdh5Cre-ERT2 mice (Dll4^iΔEC) (Dll4tm1Frad, MGI:3828266: Tg(Cdh5-cre/ERT2)1Rha, MGI:3848982)[62]. For inducible Cre-mediated recombination, mice received daily intraperitoneal (i.p.) injections of tamoxifen (4 mg/kg body weight) from day1 to day 5 (d1-d5) one week before the cranial window and calvarial drill hole lesion surgery.

For fracture healing model, we used 10-week-old female mice for the fracture experiments. Mice were anesthetized by using a ketamine hydrochloride/xylazine mixture (80 /12 mg/kg body weight, i.p.) the left leg was fractured with three-point bending. It was stabilized with an intramedullary nail (hollow needle 23 G) as described previously[81]. Carprofen (4 mg/kg intra muscular) was given as an analgesic and further on at 24-hour intervals when required. Mice were sacrificed by cervical dislocation after post-fracture surgery day at 14 and 12 weeks wild-type female were used as control.

## Bone sample preparation

Mice were sacrificed and femurs were harvested and fixed immediately in ice-cold 2% paraformaldehyde (PFA) for 6 to 8 h under gentle agitation. Bones were decalcified in 0.5 M EDTA for 16 to 24 h at 4 °C under gentle shaking agitation, which was followed by overnight incubation in cryopreservation solution (20% sucrose, 2% PVP) and embedded in in bone embedding medium (8% gelatine, 20% sucrose, 2% PVP). Samples were stored at −80 °C. For immunofluorescence staining 90 to 100 μm-thick cryosections were prepared.

## Surgical preparation for longitudinal intravital imaging

We used a chronic cranial window to provide clear optical access to the calvarial bone and bone microvessels[41,82–84]. A schematic of the cranial window construction and fixation of the mouse using the head immobilization device is shown in Fig. 1a. Mice were anesthetized with a combination of ketamine/xylazine. A circular incision was made in the scalp to expose the underlying dorsal skull surface. A dental drill (diameter: 500 μm) was used to insert a drill hole lesion into the parietal bone (Fig. 1b). A round glass cover slip with a hanging drop of sterile PBS was placed over the bone lesion and fixed with dental acrylic. A custom-designed titanium ring was glued on top to allow head immobilization and the elimination of motion artifacts. Dexamethason was used to prevent inflammatory responses. Ketamine/ xylazine anesthesia was used for intravital imaging and mice were kept on a 37 °C heat pad to keep the body temperature constant[41,82]. The physical condition of the mice was frequently controlled by observing their breathing frequency and their overall condition.

## Bone stromal cell preparation for single cell RNA sequencing (scRNA-seq)

Skulls with calvarial lesions (diameter: 1 mm) at PLD14 with their corresponding age-matched controls were harvested, cleaned from attached surrounding tissue and collected in digestion enzyme solution (Collagenase type I and IV, 2 mg/ml). Next, bones were cut into

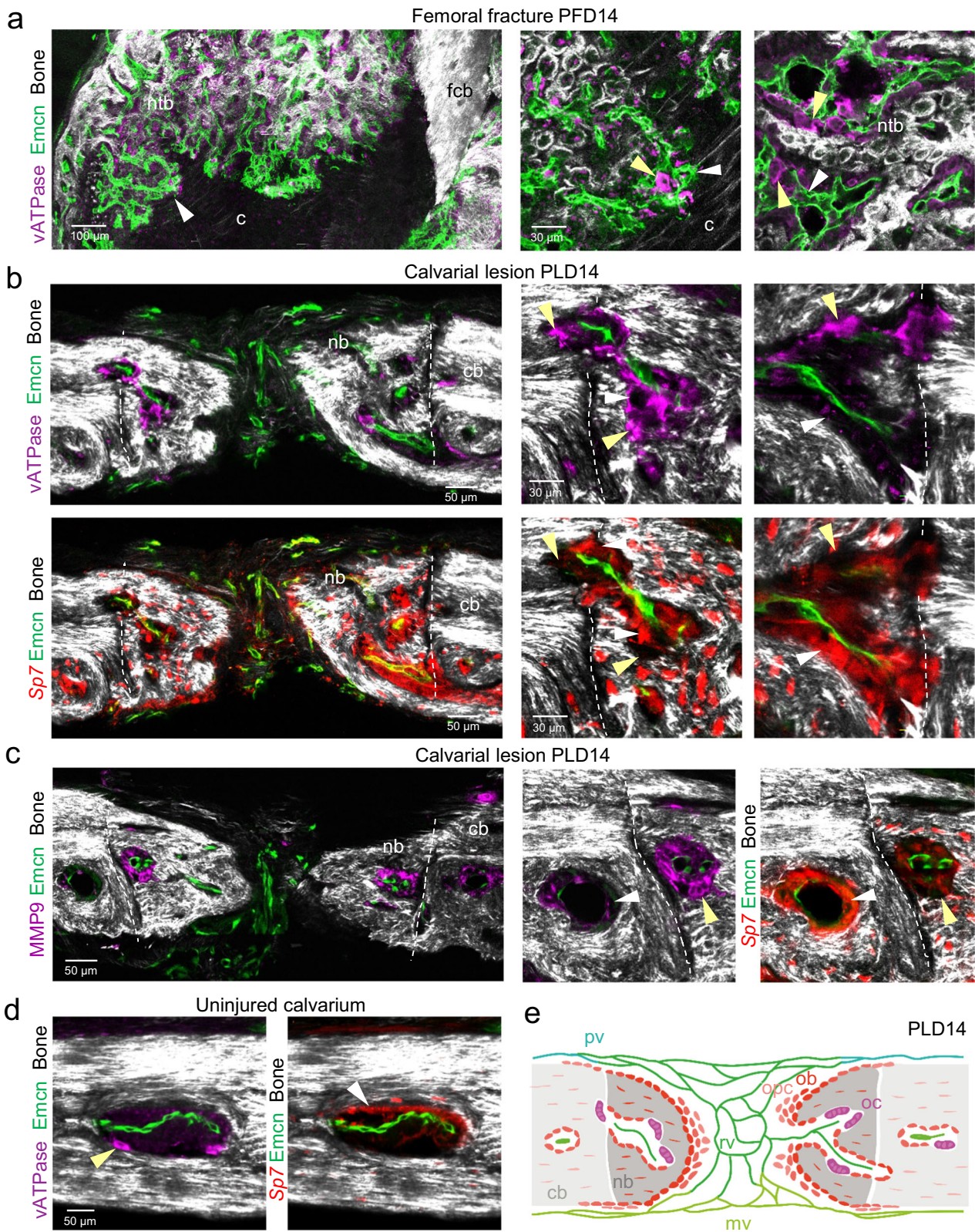

small pieces. Samples were digested for 30 min at 37 °C under gentle agitation. Digested samples were transferred to 70 μm strainers in 50 ml tubes to obtain a single–cell suspension, which was resuspended in blocking solution (1% BSA, 1 mM EDTA in PBS without Ca²⁺/Mg²⁺), centrifuged at $300 \times g$ for 5 mins, washed 2–3 times with ice-cold blocking solution, and filtered through 50 μm strainers. Pellets were resuspended in respective volume of blocking solution. Single cell

suspensions were subjected to lineage depletion using lineage cell depletion kit (MACS, cat#130-090-858) following the manufacturer's instructions. Next, lineage negative cells were depleted by CD45 and CD117 using microbeads (MACS, cat#130-052-301 and cat#130-091-224) from lineage negative (Lin) bone cells to enrich bone stromal cells. Single cell suspensions were processed with BD Rhapsody and scRNA-seq libraries were evaluated and quantified by Agilent

**Fig. 7 | Bone remodeling by osteoclasts in femoral fractures and calvarial bone lesions. a** Tile-scan multiphoton microscopy of PFD14 femoral fracture. Overview image shows maximum intensity projections of Emcn+ (green) microvasculature, vATPase+ (purple) osteoclasts and SHG+ (white) newly formed trabecular-like woven bone (ntb) and fractured compact bone (fcb). Zoom-in views show vATPase+ osteoclasts (yellow arrowhead) in close association with early Emcn+ microvessels (white arrowhead) invading the cartilage (c) at the chondro-osseous junction (middle) and vATPase+ osteoclasts in close association with remodeling Emcn+ microvessels and early SHG+ bone matrix deposition (right). **b** Multiphoton microscopy showing vATPase+ osteoclasts (top) and Sp7+ osteoblasts (bottom) together with Emcn+ microvessels residing in small cavities of newly formed SHG+ calvarial bone at PLD14. Yellow arrowheads point to vATPase+ osteoclasts located near SHG+ bone. White arrowheads point to osteoblasts located at complementary locations compared to osteoclasts. **c** Multiphoton microscopy showing MMP9+ staining and Emcn+ microvessels in calvarial bone lesion of PLD14. Note that MMP9+ staining is associated with multinucleated cells of osteoclast morphology (yellow arrowhead) located close to SHG+ bone. White arrowhead indicates Sp7+ osteoblasts located at complementary locations to MMP9 staining. **d** Multiphoton microscopy showing vATPase+ osteoclasts (left) and Sp7+ osteoblasts (right) together with Emcn+ microvessels in BM cavities of uninjured calvarial bone. **e** Schematic showing the process of intramembranous ossification after calvarial lesion injury with osteoblastic cells collectively invading the vascularized lesion tissue as a multicellular sheet. Newly formed bone cavities contain microvessels together with bone-forming osteoblasts (ob) and bone-resorbing osteoclasts at complementary locations, suggesting that cavities in newly formed bone are actively remodeling. opc osteoprogenitors, ob osteoblasts, oc osteoclasts, rv remodeling vessels, mv meningeal vessels, pv periosteal vessels. Reproducibility was ensured by n = 3 or more biologically independent experiments.

Bioanalyzer using High sensitivity DNA kit (cat#5067-4626) and Qubit (ThermoFisher Scientific, Cat# Q32851). Individual libraries were diluted to 4 nM and pooled for sequencing. Pooled libraries were sequenced by using High Output kit (150 cycle) (Illumina cat#TG-160-2002) with a NextSeq500 sequencer (Illumina).

### Single cell RNA-seq data analysis

Sequencing data of FASTQ format were processed with BD Rhapsody WTA Analysis pipeline (version 1.0) on SevenBridges Genomics online platform (SevenBridges) and expression matrix were used for further data analysis. Data normalization, dimensionality reduction and visualization were performed using Seurat (version 4.3.0) if not specified otherwise.

For initial quality control of the extracted gene-cell matrices, we filtered cells with parameters nFeature_RNA > 500 & nFeature_RNA < 6000 for number of genes per cell and percent.mito <25 for percentage of mitochondrial genes and genes with parameter min.cells = 3. Filtered matrices were normalized by LogNormalize method with scale factor=10,000. Variable genes were found using FindVariableFeatures function with parameters of selection.method = "vst", nfeatures = 2000, trimmed for the genes related to cell cycle (GO:0007049) and then used. FindIntegrationAnchors and IntegrateData with default options were used for the data integration. Statistically significant principal components were determined by JackStraw method and the first 7 principal components were used for UMAP non-linear dimensional reduction.

Unsupervised hierarchical clustering analysis was performed using FindClusters function in Seurat package. We tested different resolutions between 0.1 - 0.9 and selected the final resolution using clustree R package to decide the most stable as well as the most relevant for our previous knowledges. Cellular identity of each cluster was determined by finding cluster-specific marker genes using FindAllMarkers function with minimum fraction of cells expressing the gene over 25% (min.pct=0.25), comparing those markers to known cell type-specific genes from previous studies and further confirmed using the R package SingleR, which compares the transcriptome of each single cell to reference datasets to determine cellular identity.

For subclustering analysis, we isolated specific cluster(s) using subset function, extracted data matrix from the Seurat object using GetAssayData function and repeated the whole analysis pipeline from data normalization. FeaturePlot, VlnPlot and DoHeatmap functions of Seurat package were used for visualization of selected genes.

### Blood flow measurements

For in vivo imaging of blood flow dynamics at cellular resolution in microvessels TexasRed-dextran (70.000 MW, Molecular Probes) was injected into mice through the tail. The dextran dye labels the blood plasma, but is excluded from RBCs which appear as dark objects moving against a bright fluorescent background allowing to visualize the dynamic movements of RBCs in microvessels. Real-time movies were recorded to animate the blood flow dynamics in the calvarial lesion area, and blood flow velocities in microvessels were measured using center line scans as described previously[41,83]. Up to 25 individual vessel segments were measured in random order. To ensure flow velocities remained constant, we measured selected vessels twice at different time points, i.e. at the beginning and end of an experiment. We used a python script implemented in the acquisition software to export velocity data of single RBC streaks.

### VEGF-A plasmid construction and overexpression

To generate the pLIVE-VEGFA165-HA-MP-Asp8x bone-homing protein containing VEGF165 fused to HA-tag, metalloprotease and 8x Asp peptide sequences, a cDNA fragment encoding amino acids 1-191 of human VEGFA was amplified via PCR using the following oligonucleotide primers: VEGFA-AscI-Fwd: 5'- ATGAACTTTCTGCTGTCT-3' and VEGFA-XhoI-Rev: 5'-CCGCCTCGGCTTGTCACATCTGCA-3' and annealed with the NEBuilder Assembly Cloning Kit. 8-10-week-old *Flk1-GFP* mice were used for hydrodynamic tail vein injection. Animals were injected with 0.5 µg g⁻¹ (plasmid/body weight) pLIVE-Vegfa plasmid suspended in TransIT606 hydrodynamic delivery solution (Mirus, Cat# MIR5340). The appropriate amount of plasmid was suspended in an injection volume of 10% of the body weight and injected via the tail vein in 5-7 s as previously reported. Mice were used after 3 days for drill hole lesion injury.

### Image analysis

The ImageJ was used to analyze and quantify the newly formed bone area in the calvarial drill hole lesion. Three imaging planes at different depths (typically 50 µm, 150 µm, 250 µm) were used to calculate the newly formed bone area in the drill hole lesion. To determine the vascular area in the calvarial lesion, the maximal projection image was transformed into a binary image and the vessel area in the circular lesion area was measured. Number of vascular sprouts and branch points were analyzed by selecting three representative ROIs (280µm x 280µm) from maximum projection images (z = 20 µm), that were transformed into binary images with ImageJ to further analyze sprout numbers using AngioTool[85]. To calculate perfused vessel area, the complete vascular area of the binary images was measured and perfused vessel area was analyzed manually based on visual inspection.

### Statistics and reproducibility

Statistical analysis and design of micrographs and plots was performed with GraphPad Prism 10 Software. An unpaired two-tailed Student's t test was used for comparison between two groups. For comparison between two groups and multiple time points, two-way ANOVA with multiple comparisons was used. Data is presented as mean +/- SD. Differences were considered statistically significant at $p < 0.05$.

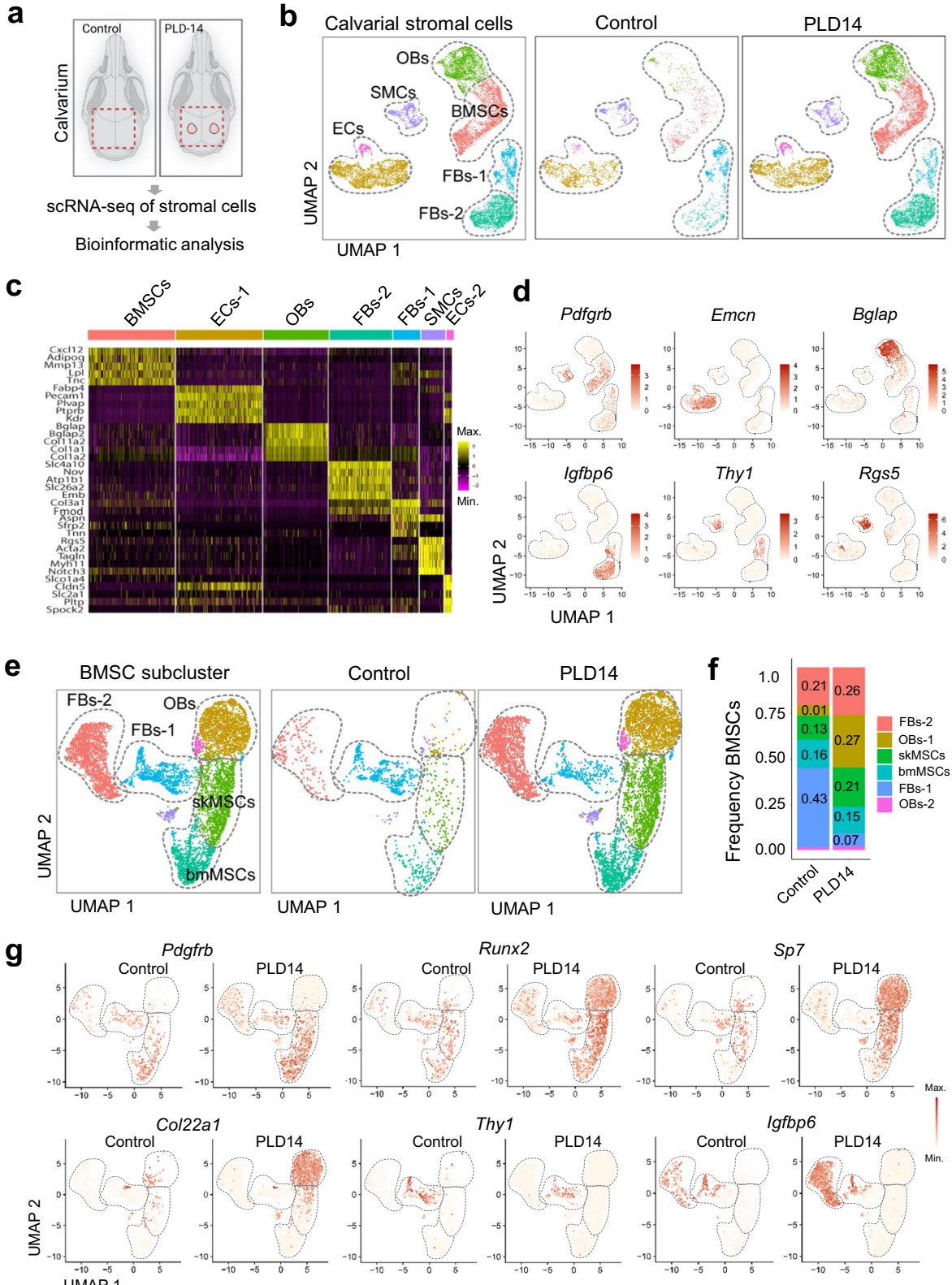

**Fig. 8 | Single cell RNA-sequencing analysis of bone non-hematopoietic cells after calvarial bone injury. a** Preparation of calvarial bones at PLD14 for scRNA-seq data analysis. **b** UMAP plots showing color-coded merged cell clusters from calvarial bones at PLD14 (left), group-selected color-coded cell clusters from controls (middle) and calvarial bone lesions (right). **c** Heat map showing top cell marker genes of each cell population shown in (**b**). **d** Cell-specific marker genes (*Pdgfrb* - BMSCs, *Emcn* - ECs, *Bglap* - OBs, *Igfbp6* - FBs-2, *Thy1* - FBs-2, *Rgs5* - SMCs) as shown in feature plots (**b**). **e** UMAP plots of BMSCs subcluster showing color-coded merged cell clusters from calvarial bones at PLD14 (left), group-selected color-coded cell clusters from controls (middle) and calvarial bone lesions (right). **f** Frequency plots of color-coded BMSC subcluster at PLD14 shown in (**e**). **g** BMSCs subcluster from control and PLD14 data displayed in UMAP plot (**e**). bmMSCs/skMSCs (*Pdgfrb, Runx2*) and osteoblasts and progenitors (*Runx2, Sp7, Col22a1*) are increased in PLD14 bone relative to control. FBs-1 (*Thy1*) are not altered and FBs-2 (*Igfbp6*) are strongly increased in PLD14 bone relative to control.

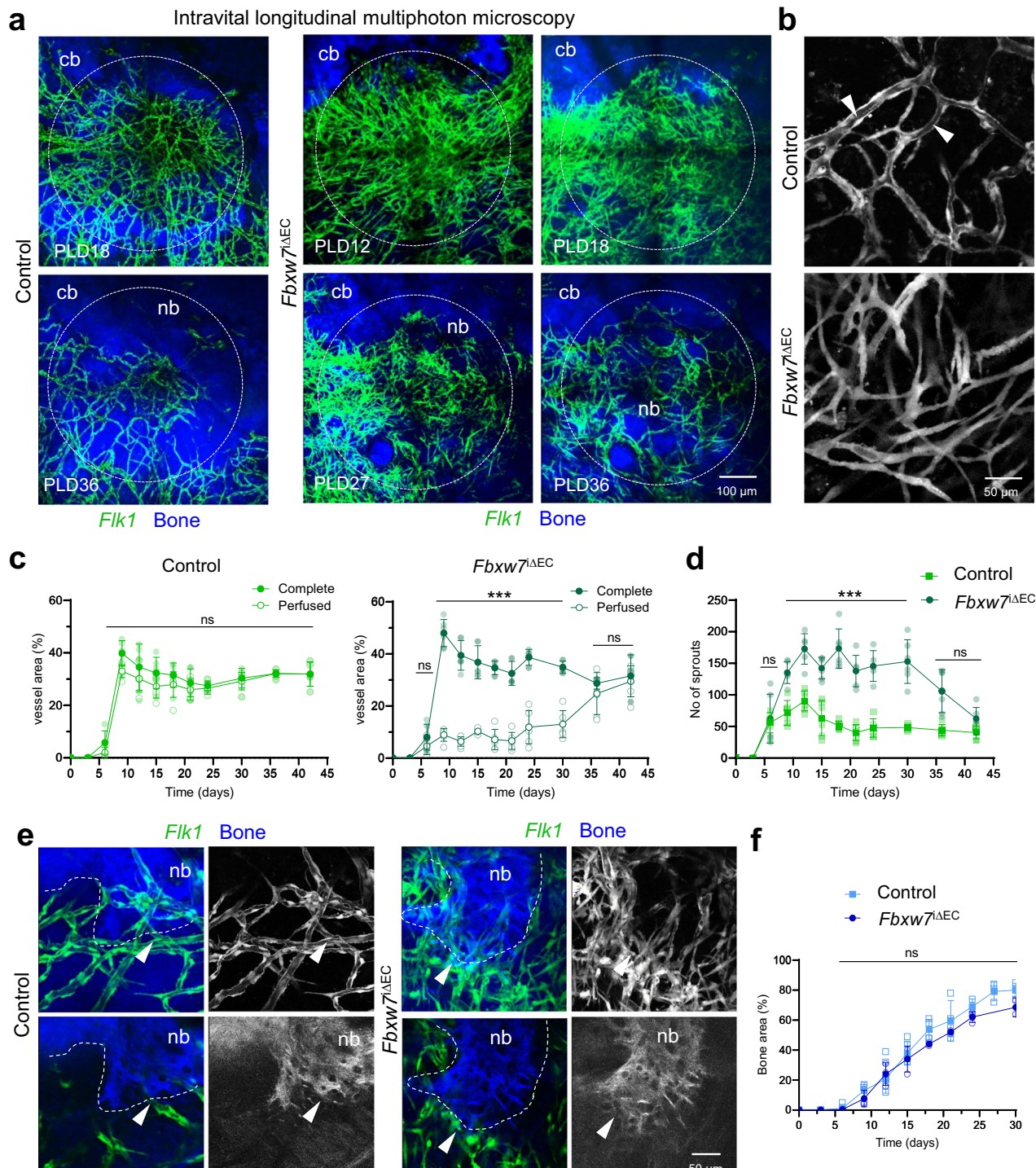

**Fig. 9 | Notch activation enhances endothelial sprouting after calvarial bone injury. a** Longitudinal intravital multiphoton microscopy showing hypersprouting of *Flk1-GFP*⁺ (green) microvessels in *Fbxw7* ^iΔEC *Flk1-GFP* transgenic mice after calvarial bone lesions at PLD12, PLD18, PLD27 and PLD36. Control *Flk1-GFP*⁺ transgenic mice are shown at PLD18 and PLD36. SHG⁺ calvarial bone (cb) and new bone (nb) are shown in blue. **b** Zoom-in view of control and *Fbxw7* ^iΔEC microvasculature of controls at PLD12. Arrowheads point to luminized vessels in control. Note that mutant microvessels are aluminal. **c** Histograms showing complete and perfused vascular areas in calvarial lesions (as 100%) in control (left) and *Fbxw7* ^iΔEC mice (right) over a period of 6 weeks. Data are presented as mean values ± SD. *n* = 6 independent vascular areas were examined over 3 biologically independent control animals and 3 biologically independent *Fbxw7* ^iΔEC animals. **d** Histograms showing the number of sprouts in the lesion area in control and

*Fbxw7* ^iΔEC mice over a period of 6 weeks. Data are presented as mean values ± SD. *n* = 6 independent vessel areas were examined over 3 biologically independent control animals and 3 biologically independent *Fbxw7* ^iΔEC animals. **e** Intravital multiphoton microscopy showing newly formed SHG⁺ bone (nb) with *Flk1-GFP*⁺ microvessels in close proximity to the growing bone edge in control and *Fbxw7* ^iΔEC mice. Maximum intensity projections (top) and single planes (bottom) are shown. Arrowheads point to microvessels in proximity to the growing bone edge in control and *Fbxw7* ^iΔEC mice. **f** Histogram showing the area of newly formed bone in calvarial lesions (as 100%) in control and *Fbxw7* ^iΔEC mice over a period of 4 weeks. Data are presented as mean values ± SD with *n* = 4 biologically independent control animals and *n* = 3 biologically independent *Fbxw7* ^iΔEC animals. Source data are provided as a Source Data file.

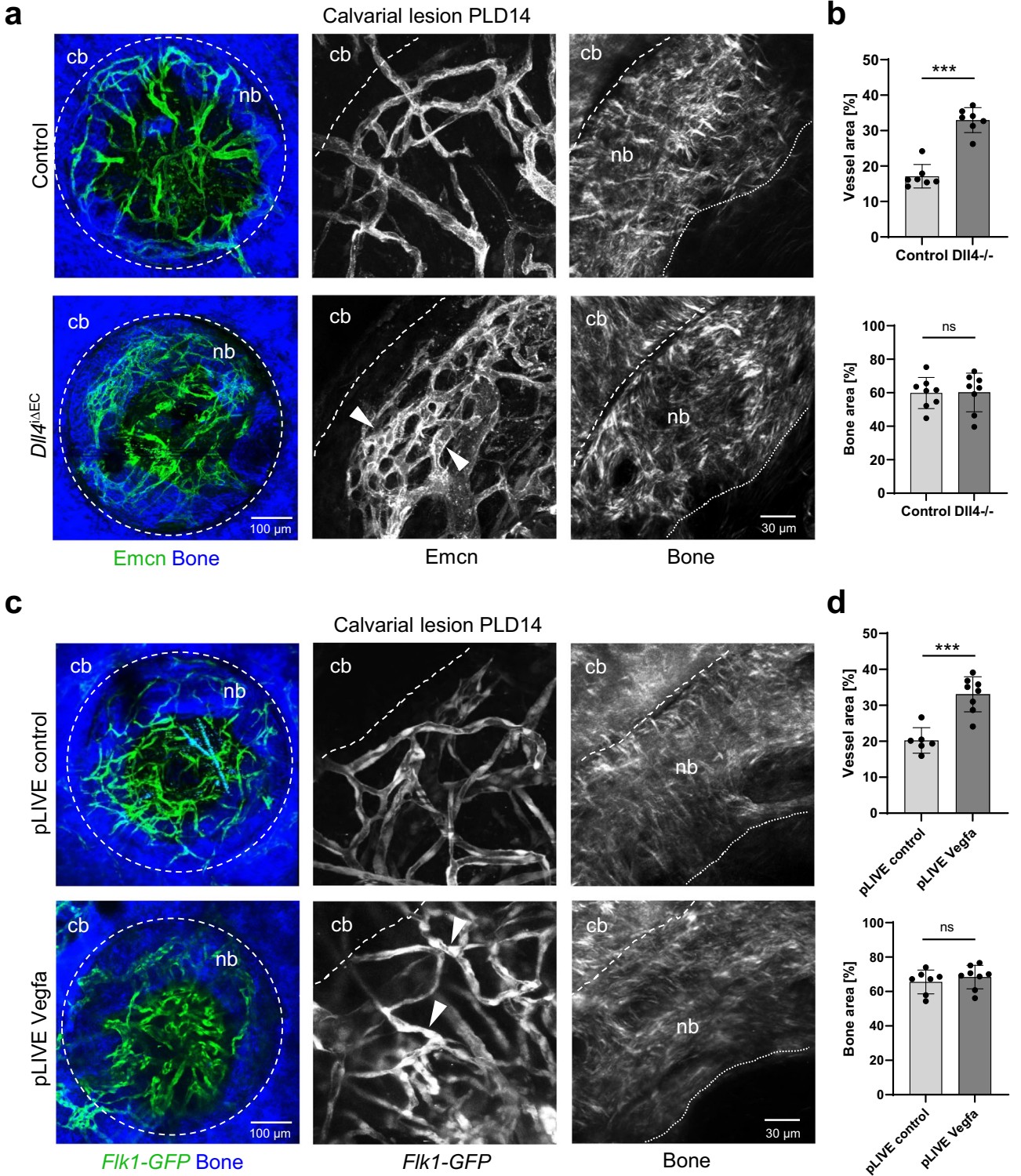

**Fig. 10 | Vascular changes via Notch or VEGF-A signaling do not affect with calvarial bone regeneration. a** Whole mount multiphoton microscopy showing Emcn⁺ (green) microvessels in *Dll4* ^iΔEC and control mice after calvarial bone lesions at PLD14. SHG⁺ calvarial bone (cb) and new bone (nb) are shown in blue. Note areas of hyperbranching and defects in vascular maturation (arrowheads). **b** Histograms showing vascular area (top) and newly formed bone are (bottom) in calvarial lesions (as 100%) in control and *Dll4* ^iΔEC mice at PLD14. Data are presented as mean values ± SD. *n* = 7 (vessel area) and *n* = 8 (bone area) independent drill holes were examined over 4 biologically independent animals. Unpaired t test, two-tailed was used for control vs. EC-specific Dll4 mutants. Vessel area p < 0.0001. Bone area *p* = 0.9494. **c** Whole mount multiphoton microscopy showing *Flk1-GFP*⁺ (green)

microvessels in mice injected with pLIVE *Vegfa* and pLIVE control three days prior to calvarial bone lesion injury, and analyzed at PLD14. SHG⁺ calvarial bone (cb) and new bone (nb) are shown in blue. Note microvessels with partially compromised vascular lumen (arrow heads). **d** Histograms showing vascular area (top) and area of newly formed bone (bottom) in calvarial lesions (as 100%) in *Flk1-GFP*⁺ mice injected with pLIVE *Vegfa* and pLIVE control at PLD14. Data are presented as mean values ± SD. *n* = 6 (vessel area) and *n* = 7 (bone area) independent drill holes were examined over 4 biologically independent animals. Unpaired t test, two-tailed was used for pLIVE control vs. pLIVE Vegfa. Vessel area *p* = 0.0001. Bone area *p* = 0.6557. Source data are provided as a Source Data file.

Reproducibility was ensured by three or more biologically independent experiments. Source data are provided as a Source Data file.

## Reporting summary

Further information on research design is available in the Nature Portfolio Reporting Summary linked to this article.

## Data availability

The single-cell RNA-seq data generated in this study have been deposited in the Gene Expression Omnibus (GEO) under accession numbers GSE154247 and GSE262350. All other data supporting the findings of this study are available from the corresponding authors on reasonable request. Source data are provided with this paper.

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

## Acknowledgements

We would like to thank M. Wewer and M. Böttcher from Miltenyi Biotec (formerly LaVision BioTec) for sharing their expertise and technical support on the TriMScope II. Funding was provided by the Max Planck Society, the University of Münster and the European Research Council (AdG 786672 PROVEC). V.M. is part of the Cells in Motion International Max Planck Research School (CiM-IMPRS).

## Author contributions

M.G.B. and R.H.A. designed experiments and interpreted results. M.G.B. generated and characterized mouse reporter and mutant lines, conducted calvarial drill hole lesion experiments and intravital multiphoton imaging experiments, their analysis and quantification, calvarial bone

sectioning and immune staining, R.S., M.T., K.K.S. and M.G.B. performed bone fracture experiments. K.K.S. performed femoral fracture sectioning, immune staining, and all single cell RNA sequencing experiments. V.M. performed calvarial bone sectioning and immune staining, A.A. performed whole-mount calvarial bone staining, S.A. cloned pLIVE plasmids. B.-I.K. performed hydrodynamic injection, M.G.B., H.-W.J. and K.K.S. analyzed the scRNA-sequencing data. K.K. programed and provided the cell viewer platform scRNA-seq analysis, M.G.B. and R.H.A. wrote the manuscript.

## Funding

## Competing interests

The authors declare no competing interests.
