## [Peer Review File · Nature Communications]

Angiogenesis is uncoupled from osteogenesis during calvarial bone regenerationREVIEWER COMMENTS

Reviewer #1 (Remarks to the Author):

General Assessment:

The manuscript entitled "Angiogenesis is uncoupled from osteogenesis during calvarial bone regeneration" provides an pretty interesting finding that by using intravital multiphoton microscopy the early vascular sprouting is not directly coupled to osteoprogenitor invasion during calvarial bone regeneration when compared with that in long bone repair. However, this study remains a phenomenological observation, and the underlying mechanistic explanation is still not clear. In addition, there is a serious logical discontinuity among the seven parts of the manuscript results, and some of the sentences are obscure and difficult to follow, making it difficult to understand the meaning quickly.

1) From the overall logic of this article, the main novelty, namely the most important finding, is the inconsistency of the vascular network with the osteoprogenitor invasion in contrast with the conventional perception that angiogenesis and osteogenesis are closely coupled in femoral fractures. Therefore, the authors should focus on building a whole story that not only demonstrating the unique vascular sprouting pattern and osteoprogenitor invasion, but also revealing the corresponding molecular mechanism.

2) Even if Notch activation was proved to enhance endothelial sprouting in calvarial bone lesions in this study, there is still a lack of direct evidence for a relationship between vessel sprouting and ossification. Data with more detailed and direct regulation mechanism is strongly suggested.

3) Figures in this manuscript are well-made with fancy multiphoton microscopy and other experimental techniques, yet results of the single-cell RNA-seq studies do not yield key information at the single-cell level and do not directly contribute to the innovative point of this article. This part is not important enough to affect the innovativeness of this study even after its removal.

Detailed comments:

1) Could the authors explain why the new bone formation starts in PLD12 in Figure1? And the result in PLD3 is suggested to be included in Figure 1c.

2) In figure 1e, how many samples in each group and each time point? As some of the values are single dotted.

3) In figure 3, could the authors explain why choose the time point PLD7, PLD9, and PLD21.

4) For figure 4e and 4f, the whole view of the pictures should be provided.

5) For figure 4 and figure 5, the authors should indicate or at least provide proof that how to exactly differ the osteoprogenitors and osteoblasts.

6) If Notch signaling dominates the vascular sprouting and has no beneficial effect on ossification in calvarial lesions, how is the effect of endothelial-specific Notch signaling KO during calvarial bone repair?

7) In discussion, other signaling pathways that promoting angiogenesis and osteogenesis rather than Notch alone should be provided?

Reviewer #2 (Remarks to the Author):

This study provides a dynamic and impressive description of vessel dynamics during bone defects. Using a drill hole model in the calvaria, coupled with multiphoton microscopy and longitudinal observations on individual animals, this study offers extensive insights into vascularization dynamics in unprecedented anatomical detail. The data are effectively supplemented with cartoons that explain the findings, as well as models of sprout extensions, tubulogenesis, and pruning, thereby describing a vascular remodeling phase upon vessel formation. Concurrent with second harmonic generation (SHG), the authors can determine the velocity of bone matrix formation alongside vessel dynamics. Additionally, the flow rate was gauged by RBC velocities, which were found to be unexpectedly low.

A striking finding was that Sp7-positive progenitor cells were not necessarily associated with blood vessels and entered the lesion primarily as groups. This contrasts with a thorough analysis of endochondral bone healing, where Emcn-positive vessels closely interact with other Emcn-positive vessels. ScRNASeq effectively supplements the analysis of sub-cell populations' composition for both calvarial and long bone lesions.

In conclusion, increased endothelial notch activity enhances vasculogenesis but hinders blood flow and slightly slows down ossification. However, it does not interfere with osteoblast differentiation processes.

This paper serves as a valuable resource for understanding the dynamic interplay between osteoblast progenitor and vessel dynamics during fracture healing.

However, a few points still need to be addressed.

Major Points:

1. Osteoclasts are posited to play a significant role in remodeling during fracture healing. They may arrive via the vessels or possibly originate from tissue resident cells. Hence, it is essential to visualize osteoclasts in this scenario, potentially in cross-sections of both calvarial and endochondrial lesions, to identify their locations relative to vessels, PDGFRb, Sp7, and Runx2-positive cells. Staining with either TRAP and/or cathepsin K is necessary to clarify this point.
2. In the early phase, Flk1-positive vessels appear to be involved in small cavities in the bone. The authors need to hypothesize why this happens. Is it a preventative measure against bone formation or a collagen resorptive process initiated by the vessels themselves or bone-resorbing cells (osteoclasts)? Do the authors anticipate MMP9 activity, as suggested in the literature? This could be resolved by conducting immunofluorescence staining on the cross-sections.

Minor points:

1. In Suppl. Fig. 2d, the alignment of FLK1-positive cells towards collagen fibers is described, but this isn't clearly evident from the figures. The images show FLK1-positive cells attached, but not necessarily aligned.
2. In Suppl. Fig. 3d, cells that resemble osteocytes (at least in their location embedded in the bone) seem to be positive for Sp7. Can the authors clarify this?
3. Can the authors confirm that PDGFRb-positive cells are negative for Sp7?

Nature Communications manuscript NCOMMS-23-21520

Angiogenesis is uncoupled from osteogenesis during calvarial bone regeneration

POINT-BY-POINT RESPONSE LETTER TO REVIEWERS

We would like to thank both reviewers for their time and effort, and their valuable suggestions, which we greatly appreciate. Based on their constructive and helpful comments, we have revised and further improved our manuscript. We provide a significant amount of new data, including additional experimental approaches to alter the regenerating vasculature during calvarial bone healing and to provide stronger evidence for the novel concept of vascular uncoupling from osteogenesis. In addition, we address the important role of osteoclasts in bone remodeling during the healing process of femoral fractures and calvarial lesions. We found that osteoclasts play an active role in the early phase of calvarial bone healing by remodeling small vascularized cavities in the newly formed bone.

We trust that the extensive revision has addressed all issues raised by the expert reviewers with the exception of the direct molecular mechanism underlying the osteo-angiogenic uncoupling. Nevertheless, to address this important comment, we have included a paragraph in the Introduction explaining that the signaling pathways regulating the co-invasion of osteoprogenitors with blood vessels are still under debate and the direct regulatory mechanism is not clear.

REVIEWER COMMENTS

Reviewer #1 (Remarks to the Author):

General Assessment:

The manuscript entitled “Angiogenesis is uncoupled from osteogenesis during calvarial bone regeneration” provides a pretty interesting finding that by using intravital multiphoton microscopy the early vascular sprouting is not directly coupled to osteoprogenitor invasion during calvarial bone regeneration when compared with that in long bone repair. However, this study remains a phenomenological observation, and the underlying mechanistic explanation is still not clear. In addition, there is a serious logical discontinuity among the seven parts of the manuscript results, and some of the sentences are obscure and difficult to follow, making it difficult to understand the meaning quickly.

1) From the overall logic of this article, the main novelty, namely the most important finding, is the inconsistency of the vascular network with the osteoprogenitor invasion in contrast with the conventional perception that angiogenesis and osteogenesis are closely coupled in femoral fractures. Therefore, the authors should focus on building a whole story that not only demonstrating the unique vascular sprouting pattern and osteoprogenitor invasion, but also revealing the corresponding molecular mechanism.

2) Even if Notch activation was proved to enhance endothelial sprouting in calvarial bone lesions in this study, there is still a lack of direct evidence for a relationship between vessel sprouting and ossification. Data with more detailed and direct regulation mechanism is strongly suggested.

3) Figures in this manuscript are well-made with fancy multiphoton microscopy and other experimental techniques, yet results of the single-cell RNA-seq studies do not yield key information at the single-cell level and do not directly contribute to the innovative point of this article. This part is not important enough to affect the innovativeness of this study even after its removal.

We thank the reviewer for this general assessment and helpful comments. Although the concept of osteogenic-angiogenic coupling is well established in long bone development and regeneration, the direct molecular mechanism underlying the co-invasion of osteoprogenitors with growing microvessels is less well understood. VEGF and hypoxia signaling components, Notch and, more recently, PDGFR β signaling have been discussed and investigated as possible mediators and signaling molecules involved in osteo-angiogenic coupling (Grosso et al. *Front. Bioeng. Biotechnol.* 2017, 5, p.68; Kusumbe et al. *Nature* 2024, 507, p.323-328; Schipani et al. *J. Bone Miner. Res.* 2009, 24, p.1347-53; Ramasamy et al. *Nature* 2014, 507, p376-380). We have added a paragraph in the Introduction and Discussion in which the current state of knowledge and the corresponding literature is mentioned.

To complicate a complex process even more, the above-mentioned factors and signaling molecules have multiple functional roles in angiogenesis and in migration and differentiation of osteoprogenitors. PDGFR β has been suggested to be a critical functional driver for mesenchymal skeletal stem cell activation, migration and angiotropism during bone repair (Bohm et al. *Dev. Cell*, 2019, 51, p.236-254). Our new data confirm previous reports that PDGFR β ⁺ cells are closely associated with growing Emcn⁺ vessels at the osseous-chondral junction of femoral fractures, but are also abundant in the early vascularized and calcified callus (**Supplementary Fig. 4a**, Sivaraj et al. *Nat. Commun.* 2022, 28, p.571). In calvarial bone lesions, PDGFR β ⁺ cells are highly abundant in the vascularized and uncalcified region (**Supplementary Fig. 4b**).

More detailed studies, beyond the scope of this manuscript, are needed to address 1) the molecular mechanism by which growing vessels are coupled to co-migrating osteoprogenitors in long bone regeneration and 2) what contributes to the uncoupling of vessels and osteoprogenitors in regenerating calvarial bone.

Detailed comments:

1) Could the authors explain why the new bone formation starts in PLD12 in Figure1? And the result in PLD3 is suggested to be included in Figure 1c.

We re-evaluated the SHG⁺ signal (bone) at PLD9 in **Fig. 1c** and adjusted the dotted line to better indicate the interface between old and new bone, and to show that new bone formation starts at PLD9. This finding is confirmed and quantified in **Fig. 1e** and also shown in **Supplementary Fig. 2c**.

All images shown in **Fig. 1c-d** were obtained by intravital imaging of live mice after cranial window surgery (**Fig. 1a**). This imaging approach does not allow visualization of early vascular sprouts (PLD3) originating from the meningeal vasculature. To study early vascular sprouting, animals were sacrificed

and whole mounts were analyzed at PLD3 and PLD6. This is a different experimental approach; therefore, we have summarized these data in **Supplementary Fig. 1b-d** and summarized the results in a schematic in **Fig. 1f**. Therefore, we would prefer to keep **Fig. 1** in its current form.

2) In figure 1e, how many samples in each group and each time point? As some of the values are single dotted.

To analyze the vascularization and the regeneration of the calvarial bone after drill hole injury, we used five mice with a chronic cranial window and imaged the mice on average every three days to follow the bone healing process. Three of the five mice were imaged until PLD42, one mouse until PLD33 and one mouse until PLD21.

The individual time points show the following sample numbers (n): PLD3-PLD15: n=5, PLD18: n=4, PLD21: n=5, PLD24: n=3, PLD27-33: n=4, PLD36, PLD42: n=3

In **Fig. 1e**, individual data points are now superimposed on the previous curves for vascular and bone regeneration. For better readability, the mean values and SD are in dark color and the individual data points are in light color. We also adjusted **Fig. 9** (former Fig. 8) accordingly and added individual data points in **Fig. 9c, 9d** and **9f**.

3) In figure 3, could the authors explain why choose the time point PLD7, PLD9, and PLD21.

At PLD7 in **Fig. 3a**, the regenerating and actively growing vasculature penetrates the lesion area and forms a primitive vascular plexus with still actively growing sprouts that extend beyond the wound edge of the calvarial bone. We chose PLD7 as an early stage to visualize the blood flow dynamics in the primitive vascular plexus and in the actively sprouting front.

The PLD9 time point in **Fig. 3d** was chosen because the regenerating vasculature has reached its maximum extension and shows an increased number of vascular connections compared to PLD7 (unpublished observations). At this stage, blood flow can be visualized in an extended and interconnected vascular network that has not yet matured into functionally mature bone vessels.

At PLD21 in **Fig. 3e**, the regenerating vasculature is more mature compared to PLD9 with a smaller vascular area and fewer vascular connections compared to PLD9 (unpublished observations) due to pruning of redundant vessels. In addition, the first blood vessels with a wider vascular lumen appear. At the same time, the newly formed bone extends significantly into the lesion site and blood vessels are often found in close proximity to the growing front. Since blood flow velocities are strongly influenced by vessel diameter, PLD21 was chosen to measure blood flow velocities and corresponding vessel diameters in different types of maturing vessels as well as in vessels in close proximity to the growing bone front.

4) For figure 4e and 4f, the whole view of the pictures should be provided.

Whole view pictures for former Fig. 4e (now **Fig. 4f**) and former Fig. 4f (now **Fig. 4g**) are shown in **Fig. 4e** and **Fig. 4g**, respectively.

5) For figure 4 and figure 5, the authors should indicate or at least provide proof that how to exactly differ the osteoprogenitors and osteoblasts.

We thank the reviewer for this comment. Osteoprogenitors and osteoblasts differ in their stage of differentiation and their ability to secrete collagen and organic bone matrix (osteoid). Collagen fibers formed by osteoblasts in their immediate vicinity can be visualized by SHG imaging. Therefore, the gradual differentiation of osteoprogenitors into osteoblasts can be visualized by the gradual formation of SHG⁺ collagen fibers around the newly differentiated osteoblasts as shown in the new **Fig. 4j** and the **Supplementary Fig. 5b**.

6) If Notch signaling dominates the vascular sprouting and has no beneficial effect on ossification in calvarial lesions, how is the effect of endothelial-specific Notch signaling KO during calvarial bone repair?

As suggested by the reviewer, we used Dll4-floxed Cdh5-CreERT mice to study the effect of loss of endothelial-specific Notch signaling during calvarial bone healing. Dll4^{iΔEC} mice show a significant increase in vascular area at PLD14 compared to controls and often regions with an abnormal vascular phenotype. The regenerating Dll4^{iΔEC} vasculature shows areas of plexus-like blood vessels that are highly interconnected and have a dilated vascular lumen. Despite the strong vascular phenotype, the progression of bone regeneration was not affected, as shown in the new **Fig. 10**.

7) In discussion, other signaling pathways that promoting angiogenesis and osteogenesis rather than Notch alone should be provided?

We thank the reviewer for this comment. We have included VEGF and hypoxia signaling components, Notch and PDGFR β signaling in the Introduction and Discussion as possible mediators and signaling pathways that are being discussed in the context of osteo-angiogenic coupling.

Reviewer #2 (Remarks to the Author):

This study provides a dynamic and impressive description of vessel dynamics during bone defects. Using a drill hole model in the calvaria, coupled with multiphoton microscopy and longitudinal observations on individual animals, this study offers extensive insights into vascularization dynamics in unprecedented anatomical detail. The data are effectively supplemented with cartoons that explain the findings, as well as models of sprout extensions, tubulogenesis, and pruning, thereby describing a vascular remodeling phase upon vessel formation. Concurrent with second harmonic generation (SHG), the authors can determine the velocity of bone matrix formation alongside vessel dynamics. Additionally, the flow rate was gauged by RBC velocities, which were found to be unexpectedly low.

A striking finding was that Sp7-positive progenitor cells were not necessarily associated with blood vessels and entered the lesion primarily as groups. This contrasts with a thorough analysis of

endochondral bone healing, where *Emcn*-positive vessels closely interact with other *Emcn*-positive vessels. ScRNASeq effectively supplements the analysis of sub-cell populations' composition for both calvarial and long bone lesions.

In conclusion, increased endothelial notch activity enhances vasculogenesis but hinders blood flow and slightly slows down ossification. However, it does not interfere with osteoblast differentiation processes.

This paper serves as a valuable resource for understanding the dynamic interplay between osteoblast progenitor and vessel dynamics during fracture healing.

However, a few points still need to be addressed.

Major Points:

1. Osteoclasts are posited to play a significant role in remodeling during fracture healing. They may arrive via the vessels or possibly originate from tissue resident cells. Hence, it is essential to visualize osteoclasts in this scenario, potentially in cross-sections of both calvarial and endochondrial lesions, to identify their locations relative to vessels, PDGFR β , Sp7, and Runx2-positive cells. Staining with either TRAP and/or cathepsin K is necessary to clarify this point.

We thank the reviewer for the constructive feedback and helpful suggestions. We have added a new paragraph on osteoclasts and their role in remodeling regenerating bone after femoral fractures and calvarial lesions (**Fig. 7a-b**). To identify osteoclasts, we stained for vATPase, a critical proton pump for bone resorption, in cross sections of regenerating femoral and calvarial bone at PFD14 and PLD14 respectively. Our stainings visualize the location of osteoclasts relative to *Emcn*⁺ vessels and Sp7⁺ osteoblastic cells in SHG⁺ regenerating bone sections (**Fig. 7b**). Furthermore, vATPase⁺ osteoclasts are visualized together with PDGFR β ⁺ and Runx2⁺ cells in femoral fractures and calvarial lesions, to assign their position relative to *Emcn*⁺ vessels and Sp7⁺ cells in SHG⁺ regenerating bone as shown in **Supplementary Fig. 4a-b**.

2. In the early phase, Flk1-positive vessels appear to be involved in small cavities in the bone. The authors need to hypothesize why this happens. Is it a preventative measure against bone formation or a collagen resorptive process initiated by the vessels themselves or bone-resorbing cells (osteoclasts)? Do the authors anticipate MMP9 activity, as suggested in the literature? This could be resolved by conducting immunofluorescence staining on the cross-sections.

We thank the reviewer for this interesting comment. We have analyzed regenerating calvarial bone at PLD14, when vascularized cavities are common in newly formed bone. We found that multinucleated vATPase⁺ osteoclasts are abundant in small cavities containing *Emcn*⁺ microvessels, where they are often located close to the bone surface. Some vATPase⁺ osteoclasts are also found close to *Emcn*⁺ microvessels. Strikingly, compared to vATPase⁺ osteoclasts, clusters of Sp7⁺ osteoblasts occupy distinct, typically mutually exclusive sites on the bone surface of vascularized cavities (**Fig. 7b**).

To determine whether vessels or bone-resorbing exhibit collagen resorptive activity, we stained calvarial bone sections at PLD14 for MMP9 expression. MMP9⁺ staining was associated with multinucleated cells of osteoclast morphology and was located in bone regions devoid of Sp7⁺ osteoblastic cells, suggesting

bone resorptive activities of osteoclasts (**Fig. 7c**).

Minor points:

1. In Suppl. Fig. 2d, the alignment of FLK1-positive cells towards collagen fibers is described, but this isn't clearly evident from the figures. The images show FLK1-positive cells attached, but not necessarily aligned.

As suggested by the reviewer, we have rephrased the sentence: Flk1-GFP⁺ vascular sprouts are typically in close proximity to or attached to SHG⁺ fibers.

2. In Suppl. Fig. 3d, cells that resemble osteocytes (at least in their location embedded in the bone) seem to be positive for Sp7. Can the authors clarify this?

In the growing bone, Sp7⁺ osteoblasts are often surrounded by the newly formed bone matrix and form early bone-embedded osteocytes that are positive for Sp7, as shown in **Fig. 5d-e** and **Supplementary Fig. 4c-e**.

3. Can the authors confirm that PDGFRb-positive cells are negative for Sp7?

We observe that Sp7⁺ osteoblastic cells lining the front of the growing bone and the bone surfaces of small BM-like cavities and PDGFRβ⁺ stromal cells found in the uncalcified lesion site and occasionally in bone cavities appear to occupy complementary regions within the calvarial bone lesion (**Supplementary Fig. 4c-e**). Therefore, we can confirm that PDGFRβ cells do not express Sp7⁺.

REVIEWERS' COMMENTS

Reviewer #1 (Remarks to the Author):

- 1) Even if it is beyond the scope of this manuscript, the biggest challenge, which is also stated by the authors, is the shortage of knowledge and the corresponding molecular mechanisms behind the osteo-angiogenic uncoupling correlation during calvarial bone regeneration.
- 2) The authors do provide amount of new data on deciphering the relationship between the vessel sprouting and ossification either by notch signaling cKO model or systemic pLIVE vector injection. Could the authors help explain why the vasculature became hyperbranching though the osteogenesis was unaffected. Also, why choose a systemic pLIVE vector which allows constitutive protein expression in the liver after hydrodynamic tail vein injection, instead of local delivery of VEGF-A?
- 3) As we raised and insisted before, the results of single-cell RNA-seq studies in Figure 8 do not directly contribute to the innovation or prerequisite data in this manuscript. so it is strongly recommended to be removed or place it in the supplementary materials for improving the logic and readability of the manuscript.
- 4) In figure 3e, vessels were categorized according to the blood flow velocities; however, it leads to the final classification of sinusoidal capillaries being thicker than arteries. Is that reasonable and why?
- 5) Still, there is an inconsistency of time points among figures, as PLD6 shown in Figure 2 and 4, while PLD7 was chosen to demonstrate the vessel blood flow. As the vessels were already found as early as in PLD6.

Reviewer #2 (Remarks to the Author):

The authors addressed my concerns. I congratulate them for this exciting interesting project.

Nature Communications manuscript NCOMMS-23-21520A

Angiogenesis is uncoupled from osteogenesis during calvarial bone regeneration

POINT-BY-POINT RESPONSE LETTER TO REVIEWER #1'S COMMENTS

We would like to thank both reviewers again for their time and effort and for their valuable suggestions, which we greatly appreciate.

Please find below our detailed response to Reviewer #1's final comments. We believe that the final revision has addressed all of the remaining issues and we thank him/her for raising these additional comments.

REVIEWERS' COMMENTS

1) Even if it is beyond the scope of this manuscript, the biggest challenge, which is also stated by the authors, is the shortage of knowledge and the corresponding molecular mechanisms behind the osteo-angiogenic uncoupling correlation during calvarial bone regeneration.

We appreciate the reviewer's comment, but confirm that it's outside the scope of this manuscript. However, it would be very interesting to follow up in a further study to understand what triggers and regulates the uncoupling of co-migrating osteoprogenitors from the adjacent expanding vascular network.

2) The authors do provide amount of new data on deciphering the relationship between the vessel sprouting and ossification either by notch signaling cKO model or systemic pLIVE vector injection. Could the authors help explain why the vasculature became hyperbranching though the osteogenesis was unaffected.

Notch is a key regulator of endothelial cell fate and important for vascular differentiation and specialization. Moreover, Notch promotes the distinction between tip and stalk cells in newly forming sprouts, and is therefore critical for the formation of new sprouts and vascular connections. cKO models that affect Notch signaling result in impaired vascular sprout formation, including phenotypes of excessive sprouting and hyperbranching. Vegfa, overexpressed by pLIVE vector injection, acts upstream of Notch signaling in endothelial cells by binding to VEGFR2, inducing Dll4 expression in tip cells and Notch signaling in adjacent stalk cells.

Osteogenesis is most likely unaffected by vascular hypersprouting, because regenerating blood vessels are uncoupled from co-migrating osteoprogenitors. New bone formation is initiated at a stage when the calvarial lesion is already fully vascularized.

Also, why choose a systemic pLIVE vector which allows constitutive protein expression in the liver after hydrodynamic tail vein injection, instead of local delivery of VEGF-A?

For a gain-of-function experiment, we used cDNA encoding a bone-homing version of Vegfa in the pLIVE vector. The systemic approach allows continuous and stable overexpression of Vegfa throughout the

entire regeneration process until PLD14. If Vegfa were administered locally at the time of lesion injury (PLD0), continuous and stable Vegfa delivery would not be possible. Instead, local Vegfa levels would gradually decrease over time. To compensate for declining Vegfa concentrations, repeated applications of Vegfa would have to be used. This approach would result in repeated and unwanted irritation of the regenerating bone tissue due to multiple applications.

3) As we raised and insisted before, the results of single-cell RNA-seq studies in Figure 8 do not directly contribute to the innovation or prerequisite data in this manuscript. So it is strongly recommended to be removed or place it in the supplementary materials for improving the logic and readability of the manuscript.

The scRNA-seq data were not altered.

4) In figure 3e, vessels were categorized according to the blood flow velocities; however, it leads to the final classification of sinusoidal capillaries being thicker than arteries. Is that reasonable and why?

Blood vessels in the bone marrow of long bones and calvarial bones form a highly specialized vasculature with thin arterial vessels connecting to a complex and irregular network of downstream sinusoidal capillaries with a much wider lumen (Sivaraj et al. 2016, *Development*143, p.2706-2715). In calvarial bone marrow, arterial vessels with diameters of $8\pm 1\ \mu\text{m}$ show blood flow velocities of $1.95\pm 0.57\ \text{mm/s}$, while sinusoidal capillaries are wider with diameters of $21.1\pm 10.7\ \mu\text{m}$ and blood flow velocities of $0.23\pm 0.22\ \text{mm/s}$ (Bixel et al. 2017, *Cell Reports* 18, p.1804-1816) The enlarged lumen of sinusoidal capillaries allows for slower blood flow and significantly reduced wall shear stress, allowing for the homing of hematopoietic stem and progenitor cells and the trafficking of immune cells to and from the bone marrow compartment.

5) Still, there is an inconsistency of time points among figures, as PLD6 shown in Figure 2 and 4, while PLD7 was chosen to demonstrate the vessel blood flow. As the vessels were already found as early as in PLD6.

The PLD7 stage was chosen for blood flow measurement primarily for technical reasons. At PLD6, early sprouting vessels of the expanding vascular network are not always reliably close enough to the outer wound edge to allow the high-resolution imaging of blood flow blood with cellular resolution of moving blood cells and visualization of endothelial filopodia extensions. If TexasRed dextran is injected at a too early stage, when the vascular sprouts are still deep in the calvarial lesion, the recorded movie quality of the blood flow is not very good. Blood flow imaging cannot be repeated in the same animal at a later time point, i.e., the following day at PLD7, because much of the injected TexasRed dextran has leaked out of the vessels into the surrounding tissue within 1-2 hours, dramatically increasing the background labeling. In addition, macrophages in the vicinity of these vessels have taken up the surrounding TexasRed dextran, brightly labeling these cells. Animals can be imaged again tentatively one week after TexasRed dextran injection, when most of the label has been cleared. For these reasons, we chose PLD7 over PLD6 to image early blood flow in the regenerating calvarial vasculature.